# Convergent evidence for the temperature-dependent emergence of silicification in terrestrial plants

Zhihao Pang [1], Félix de Tombeur [2,3], Sue E. Hartley[4], Constantin M. Zohner [5], Miroslav Nikolic [6], Cyrille Violle [2], Lidong Mo [5], Thomas W. Crowther [5], Dong-Xing Guan[1], Zhongkui Luo [1], Yong-Guan Zhu [7], Yuxiao Wang[8], Ping Zhang [1], Hongyun Peng[1], Caroline A. E. Strömberg [9], Nina Nikolic [6] ✉ & Yongchao Liang [1] ✉

Research on silicon (Si) biogeochemistry and its beneficial effects for plants has received significant attention over several decades, but the reasons for the emergence of high-Si plants remain unclear. Here, we combine experimentation, field studies and analysis of existing databases to test the role of temperature on the expression and emergence of silicification in terrestrial plants. We first show that Si is beneficial for rice under high temperature (40 °C), but harmful under low temperature (0 °C), whilst a 2 °C increase results in a 37% increase in leaf Si concentrations. We then find that, globally, the average distribution temperature of high-Si plant clades is 1.2 °C higher than that of low-Si clades. Across China, leaf Si concentrations increase with temperature in high-Si plants (wheat and rice), but not in low-Si plants (weeping willow and winter jasmine). From an evolutionary perspective, 77% of high-Si families (>10 mg Si g⁻¹ DW) originate during warming episodes, while 86% of low-Si families (<1 mg Si g⁻¹ DW) originate during cooling episodes. On average, Earth's temperature during the emergence of high-Si families is 3 °C higher than that of low-Si families. Taken together, our evidence suggests that plant Si variation is closely related to global and long-term climate change.

More than half of present-day terrestrial net primary productivity is produced by plants that actively accumulate silicon (Si)[1]. Plant silicification – that is, the deposition of hydrated amorphous silica between and within cells – has received increasing attention over the last two decades, with a particular focus on its role in plant defense against herbivores[2,3] and in alleviating various environmental stresses[4,5]. Although the role and functions of Si in plant biology have been extensively studied[6,7], the reasons for the up to hundred-fold variation in plant Si concentrations[8,9] and for the emergence of high-Si species over the course of evolution remain unclear[10–12]. Understanding this

¹Ministry of Education Key Laboratory of Environment Remediation and Ecological Health, College of Environmental & Resource Sciences, Zhejiang University, Hangzhou 310058, China. ²CEFE, Univ Montpellier, CNRS, EPHE, IRD, Montpellier, France. ³School of Biological Sciences and Institute of Agriculture, The University of Western Australia, Perth, WA, Australia. ⁴School of Biosciences, University of Sheffield, Sheffield, UK. ⁵Institute of Integrative Biology, ETH Zurich (Swiss Federal Institute of Technology), Zurich 8092, Switzerland. ⁶Institute for Multidisciplinary Research, University of Belgrade, Kneza Viseslava 1, 11030 Belgrade, Serbia. ⁷State Key Laboratory of Urban and Regional Ecology, Research Center for Eco-Environmental Sciences, Chinese Academy of Sciences, Beijing 100085, China. ⁸State Key Laboratory of Plant Physiology and Biochemistry, College of Life Sciences, Zhejiang University, Hangzhou 310027, China. ⁹Department of Biology, University of Washington, Seattle, WA 98195, USA. ✉e-mail: nina@imsi.bg.ac.rs; ycliang@zju.edu.cn

phenomenon is key since evolutionary phases of plant silicification might have impacted Si cycling and its coupling with the carbon (C) cycle on different time and spatial scales[13].

The discovery of genes coding the expression of proteins facilitating Si uptake from soils and redistribution in plants provided some explanations for the variation in leaf Si concentrations[14–16]. Research has shown that the accumulation of Si is closely related to the expression levels, localization, and polarity of these transport proteins in different plant species[17]. These discoveries provided a molecular basis for our understanding of Si accumulation, but did not tell us why and when silicification emerged in plants. Many different hypotheses have been put forward (e.g., increased silicification through co-evolution with mammalian grazers, substitution of C by Si during periods of low atmospheric $CO_2$ levels, adapting to stressful habitats with seasonal aridity, low nutrient availability, or strong wind exposure)[11,12,18], but linking the emergence of high-Si taxa with one single environmental factor is not well supported by chronology[11].

One environmental factor that could have influenced the emergence of high-Si taxa is temperature[19]. The deposition of Si in the cell walls of leaf epidermis not only enhances the thermal stability of cell membranes and cell walls, but also cools the leaves through efficient mid-infrared heat emission[20,21]. As such, Si could play a key role in helping plants resist high temperature stress[22], and therefore plant silicification could have been promoted through natural selection during warm geological periods. Furthermore, lab experiments show that Si is highly plastic in response to increased temperature[19,23,24]. While this plasticity could be attributed to increased transpiration accelerating passive Si absorption[25,26], it could also highlight a positive role of Si against heat stress[20]. Although interactions between temperature and Si have been suggested, we know very little about the role of long-term temperature shifts in the emergence of high-Si taxa. Yet, understanding whether the differentiation of high- and low-Si species and historical climate change are linked is essential to better understand plant Si variation, and hence the role of vegetation in global biogeochemical cycles.

To investigate relationships between long-term temperature changes and plant silicification, we integrated evidence from experiments, field studies and existing databases (see Supplementary Fig. S1 for details about the databases). We first investigated the effect of Si on rice, a model Si-accumulating species, when grown under high and low temperature stress, and tested the effect of temperature increase on leaf Si concentration of rice, disentangling the roles of temperature and transpiration by controlling humidity. We then analyzed whether the global distribution of present-day high- and low-Si clades is related to temperatures in different regions. We also used samples from high-Si (wheat and rice) and low-Si (weeping willow and winter jasmine) species collected along climatic gradients across China, to test Si intraspecific variation in responses to air temperature. Finally, we analyzed the relationship between the Earth's historical temperature and the periods when high- and low-Si families differentiated.

## Results

### Effect of Si on rice resistance to heat and cold stress
Our first experiment tested the effect of Si on heat, cold and freeze-thaw cycling stress in an emblematic Si-accumulating species, namely rice (*Oryza sativa* L. spp. *japonica*, var Nipponbare) (see Materials and Methods for details). After planting rice in nutrient solutions containing 1.0 mM silicic acid (+Si) or not (-Si) under 30 °C for 15 days, no differences in shoot length, root length, and chlorophyll concentrations between the -Si and +Si groups were observed (five replicates, Supplementary Table S1). However, when rice plants were placed under high temperature (40 °C), low temperature (0 °C), or freeze thawing (−2 °C for 12 h and 2 °C for 12 h each day) stress for three days, a pronounced effect of Si was observed (Fig. 1a). Under heat stress, Si-fertilized rice had significantly higher shoot length, root length, and fresh weight than the -Si group (Fig. 1b–e). In contrast, root length and

fresh weight of the +Si group were lower than those of the -Si group under cold stress (Fig. 1). Under freeze thawing, the +Si group even showed significant wilting, and the shoot length, root length, fresh weight, and chlorophyll concentration of the +Si group all showed significant decreases compared to the -Si group (Fig. 1).

In addition, in order to investigate the effects of Si on heat and cold stress in rice varieties with different water requirements, we also tested one type of lowland rice, usually grown submerged, and three types of upland rain-fed rice (including *indica*, cv. Jinzao 47 [lowland rice], *japonica*, cv. Linhan 1 [upland rice], *japonica*, cv. Danhan 53 [upland rice], and *japonica*, cv. Zhenghan 10 [upland rice]). We found that added Si promoted the growth in all these varieties under high temperatures (40 °C), significantly increasing shoot length, root length, fresh weight, and chlorophyll concentration. However, added Si was harmful to them under low temperature (0 °C), manifested by more severe wilting and 8–46% decreases in shoot length, root length, fresh weight and chlorophyll concentration (Supplementary Fig. S2).

### Effect of temperature on Si accumulation in rice
Our second experiment tested the effect of temperature on Si accumulation in rice. We placed rice seedlings in 1/4-strength Kimura B solution containing 1.0 mM silicic acid at two temperatures (26 °C and 28 °C, considering that the optimal growth temperature range of rice is 25–30 °C[27]), and used humidifiers to control the humidity of the incubator at 70%. After 10 days, we measured leaf concentrations of Si, phosphorus, potassium, calcium, manganese, magnesium, copper, and zinc in the two groups. We found that, compared with the rice grown at 26 °C, a temperature increase of 2 °C resulted in a 37% increase in leaf Si concentrations ($p < 0.001$, Supplementary Fig. S3). The increase in temperature had no significant effect on other elements, except for manganese (Supplementary Fig. S3).

### Distribution of high- and low-Si clades and temperature
Following these experiments, we used information on species distribution obtained from the GBIF (Global Biodiversity Information Facility) to test whether high-Si and low-Si plant clades were associated with different temperatures. From Hodson et al., we selected the top ten plant clades with the highest Si concentrations (high-Si clades) and the bottom ten clades with the lowest Si concentrations (low-Si clades). Specifically, the high-Si clades were Poales, Saxifragales, Schisandraceae, Arecales, Boraginaceae, Fagales, Rosales, Nymphaea-ceae, Asterales, and Alismatales. The low-Si clades were Acorales, Brassicales, Liliales, Asparagales, Aquifoliales, Santalales, Pandanales, Celastraceae, Cornales, and Bromeliaceae. We randomly selected the distribution information of 1000 occurrences from each clade using rgbif package in R (Supplementary Code 1, Source Data), and drew distribution maps of high-Si and low-Si plant clades.

Our analysis suggests that high-Si plants occur to a greater extent in areas of lower latitude while low-Si plants are more noticeably present in higher latitude regions (Fig. 2a). Comparing the mean annual air temperatures (MAT) of high-Si and low-Si plant distribution areas, we found that the average temperature of the distributions of high-Si plants is 1.2 °C higher than that of low-Si plants (Fig. 2b, two-sided Wilcoxon rank sum test, W = 56105401, $p < 2.2e-16$), and the median and mode are 1.6 °C and 6.0 °C higher than those of low-Si plants, respectively (Fig. 2c).

### Leaf Si concentration and climate variables
Along with global analyses, we conducted field sampling of two typical high-Si species (wheat [*Triticum aestivum* L., Poaceae] and rice [*Oryza sativa* L., Poaceae]) and two low-Si species (weeping willow [*Salix babylonica* L., Salicaceae] and winter jasmine [*Jasminum nudiflorum* Lindl., Oleaceae]) across China (Fig. 3a, Source Data). We found a significant positive correlation between wheat leaf phytolith (extracted by microwave digestion method, followed by drying and weighing; see

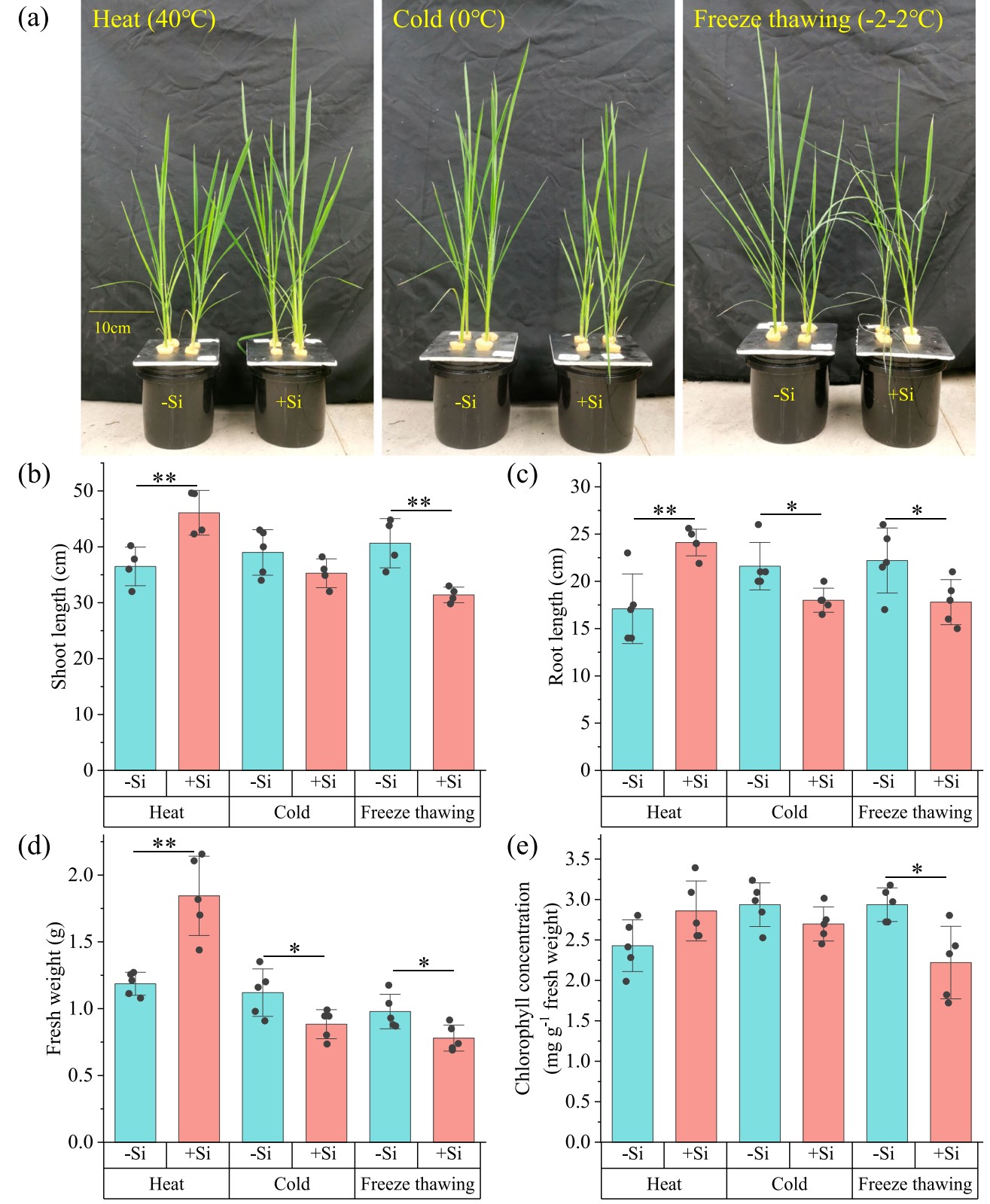

**Fig. 1 | The effect of Si on rice growth under high temperature, low temperature, and freeze thawing stress.** Images of rice (**a**) and shoot length (**b**), root length (**c**), fresh weight (**d**) and chlorophyll concentration (**e**) of rice after three days of under high temperature (40 °C), low temperature (0 °C), or freeze thawing (−2 °C for 12 h and 2 °C for 12 h each day) stress. The +Si group refers to the cultivation of 20-day old rice in a nutrient solution containing 1.0 mM silicic acid for 15 days prior to stress. The data represent the mean and standard deviation of five replicates. Differences between groups are compared using two-sided Welch t-tests (the *p* values are displayed in Source Data). Significant differences between the groups without and with Si are indicated as follows: *$p < 0.05$, **$p < 0.01$, ***$p < 0.001$.

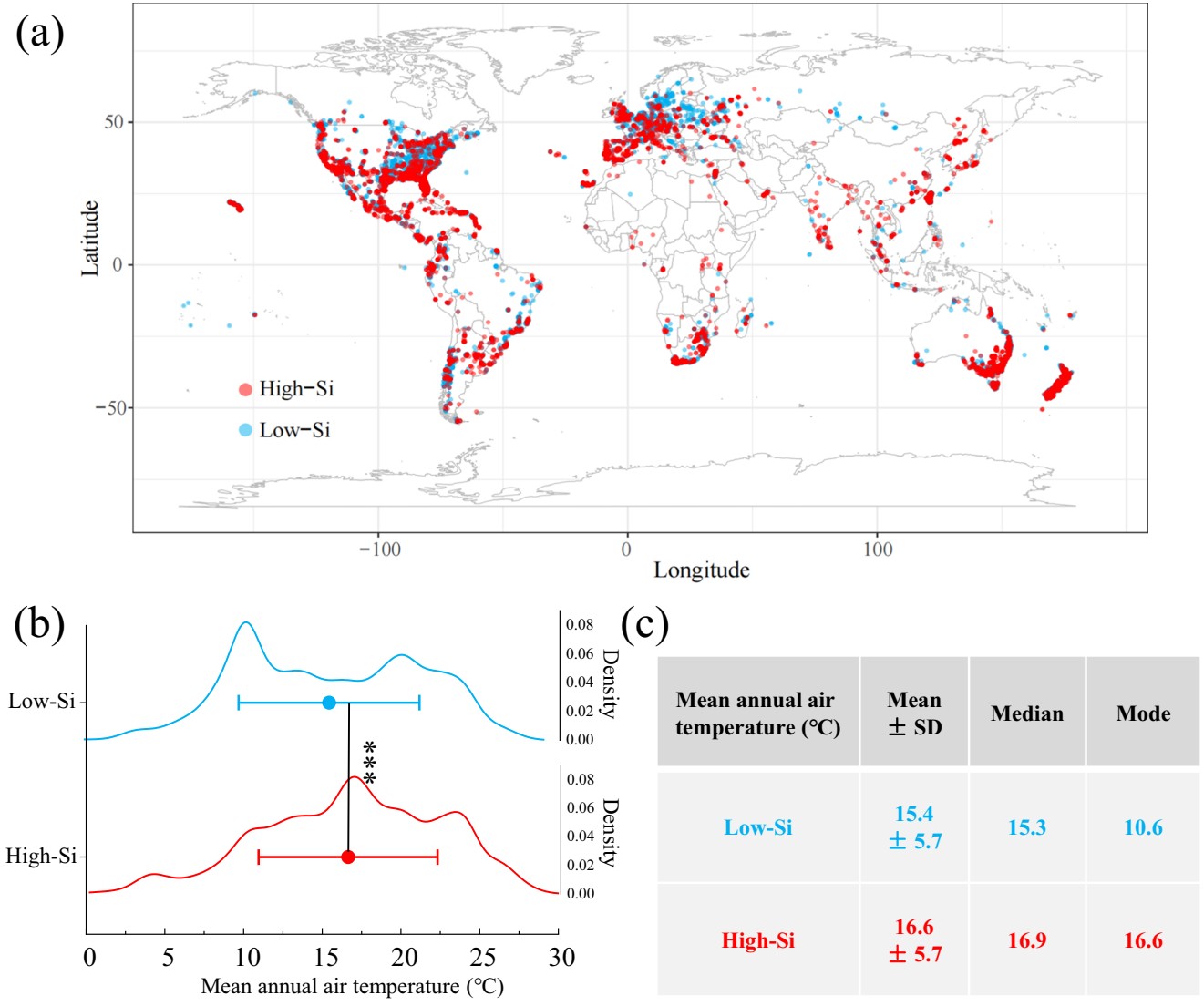

**Fig. 2 | Relationship between the distribution of high- and low-Si plant clades and temperature. a** Global distribution of sampling sites for high- ($n = 9972$ occurrences, red dots) and low-Si ($n = 9990$ occurrences, blue dots) plant clades. Data was obtained from GBIF using rgbif package in R. **b** Temperature range of high- ($n = 9972$ occurrences) and low-Si ($n = 9990$ occurrences) plants distributions. The mean annual air temperatures were obtained from CHELSA (chelsa-climate.org). Data are presented as the mean and standard deviation. Differences between groups are compared using two-sided Wilcoxon rank sum test ($p < 2.2e$ $-16$). Significant differences between the low-Si and high-Si plants are indicated as follows: ***$p < 0.001$. **c** The mean ± SD, median, and mode of distribution temperature of high- and low-Si plants.

methods) concentrations and MAT ($R^2 = 0.18$, $p < 0.001$) and a significant but weak positive correlation between rice leaf phytolith concentrations and MAT ($R^2 = 0.03$, $p < 0.01$) (Fig. 3b, Supplementary Code 2). Additionally, we conducted a separate analysis for upland rice to control for differences in rice planting methods[28]. Compared with lowland rice, the results showed that there was a stronger positive correlation between leaf phytolith concentration and MAT in upland rice (Supplementary Fig. S4, $R^2 = 0.23$, $p < 0.01$). In the two low-Si species, correlations between Si concentrations (measured by ICP-OES due to low Si content; see methods) and MAT were not significant (Fig. 3b, for weeping willow $R^2 = 0.001$, $p = 0.75$, for winter jasmine $R^2 = 0.006$, $p = 0.49$).

We also collected eight other climatic variables (see Source Data), including mean ground surface temperature (GST), evaporation (EVP), precipitation (PRE), pressure (PRS), relative humidity (RHU), wind speed (WIN), surface downwelling shortwave flux in air (RSDS), and vapor pressure deficit (VPD). All variables were standardized to a mean of 0 and SD of 1. We then constructed multiple regression models using ordinary least squares (OLS) to evaluate the relative importance of each climatic variable in explaining changes in leaf phytolith concentration of high-Si plants[29].

The results indicated that, for both wheat and rice, MAT is the most important positive influencing factor explaining the concentration of leaf phytoliths (Fig. 3c, d, Supplementary Table S2). In addition, in the regression model calculated using the relaimplo package in R, the contribution of MAT to the leaf phytolith concentration of wheat and rice ranked first among many climatic variables (Fig. 3e, f, Supplementary Code 2). Overall, our results suggest that present-day temperature influences both the distribution of high- and low-Si plant clades worldwide, and also affects intraspecific variation in leaf Si concentrations of high-Si species.

## Evolution of plant silicification and Earth temperature

We then constructed evolutionary trees of Si transporters proteins (encoded by the genes *Lsi1*, *Lsi2*, *Lsi3* and *Lsi6*)[16]. We searched for the

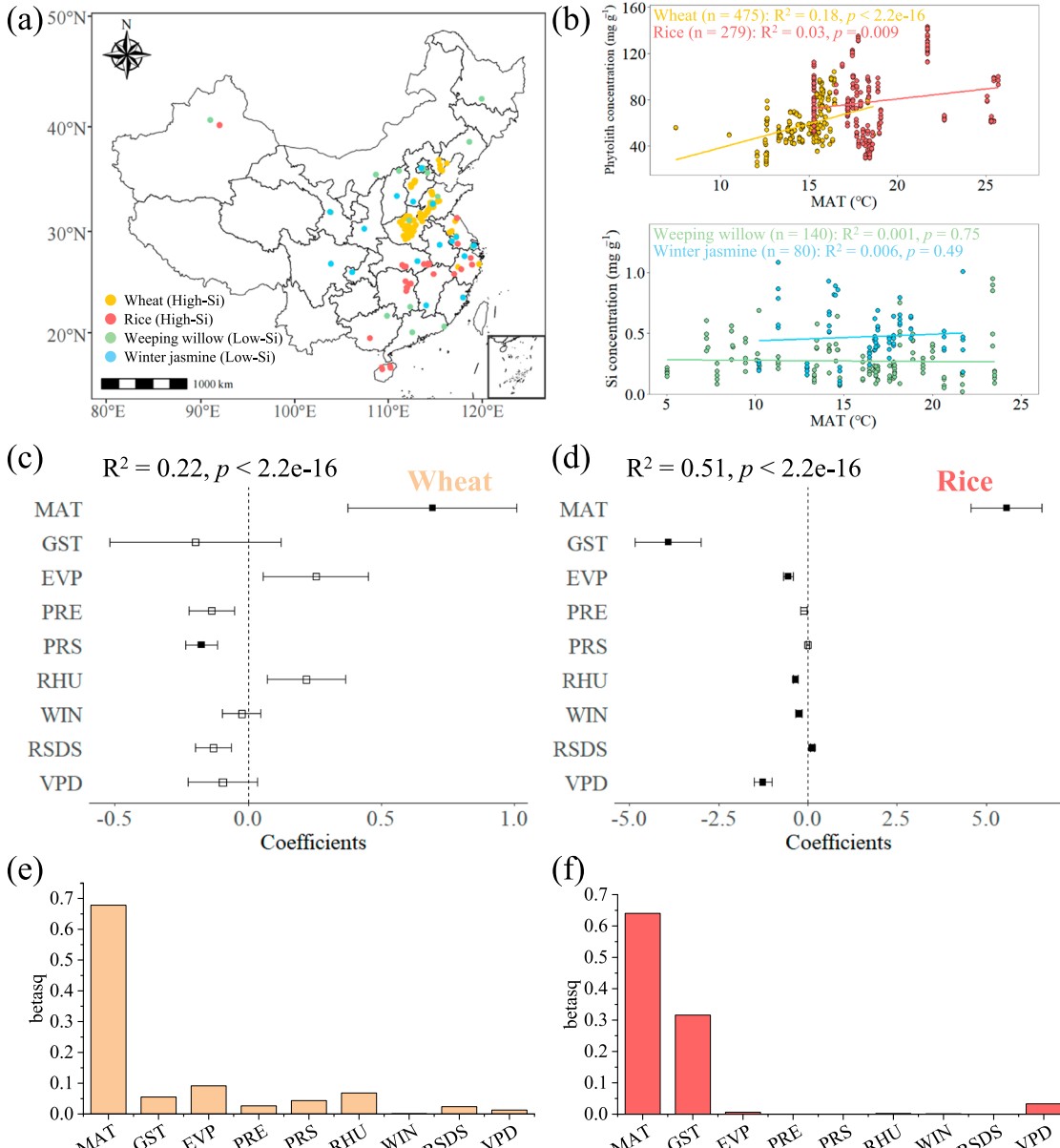

**Fig. 3 | Field sampling of typical high-Si (wheat and rice) and low-Si (weeping willow and winter jasmine) plants in China. a** The source of sampling for plant samples (wheat, $n = 475$ individuals; rice, $n = 279$ individuals; weeping willow, $n = 140$ individuals; winter jasmine, $n = 80$ individuals). The Chinese map data is sourced from DATAV.GeoAltas (http://datav.aliyun.com/portal/school/atlas/area_selector). **b** Correlation of phytolith/Si concentration with mean annual temperature. The Student's $t$ test for the regression coefficient is two-sided. **c, d** Coefficients of the multiple regression analysis of climatic variables and phytolith concentration in wheat ($n = 475$ individuals) and rice ($n = 279$ individuals) leaves. The climatic variables include mean annual air temperature (MAT), ground surface temperature (GST), evaporation (EVP), precipitation (PRE), pressure (PRS), relative humidity (RHU), wind speed (WIN), surface downwelling shortwave flux in air (RSDS), and vapor pressure deficit (VPD). The Student's $t$ test for the regression coefficient is two-sided, with no adjustments for multiple comparisons. The error bars represent the standard error of the regression coefficients. Solid rectangles indicate significant correlation ($p < 0.05$), while hollow rectangles indicate insignificant correlation. **e, f** The relative importance of various climate variables in explaining the phytolith concentration in wheat and rice leaves assessed by squared standardized coefficients (betasq).

homologous protein sequences of the top 20 species with highest similarity to that of rice (*Oryza sativa* ssp. *Japonica*, the species where these transporters proteins were first discovered) using BLASTP (Basic Local Alignment Search Tools for Proteins)[30] for evolutionary tree construction (see Supplementary Data 1 for the sequence information). We used MEGA11[31] to construct a Maximum-Likelihood phylogenetic tree and ProtTest[32] to select the most suitable amino acid substitution model. Then we used Phylogenetic Analysis by Maximum Likelihood (PAML) to construct the corresponding timetree[33], estimated the differentiation time of Lsi

proteins using MCMCtree[34], and checked the paired differentiation time of three couples of species provided by Timetree (https://timetree.org/, see Supplementary Table S3 for the pairwise divergence time used to construct the timetree for each protein)[35]. The results showed that the Si transporter proteins of species with higher Si concentrations evolved mainly during warm periods (Supplementary Fig. S5). However, due to the high conservation of the Lsi series proteins, homologous proteins produced by the BLASTP using the rice Lsi proteins mostly belong to Poaceae, which makes the analysis limited.

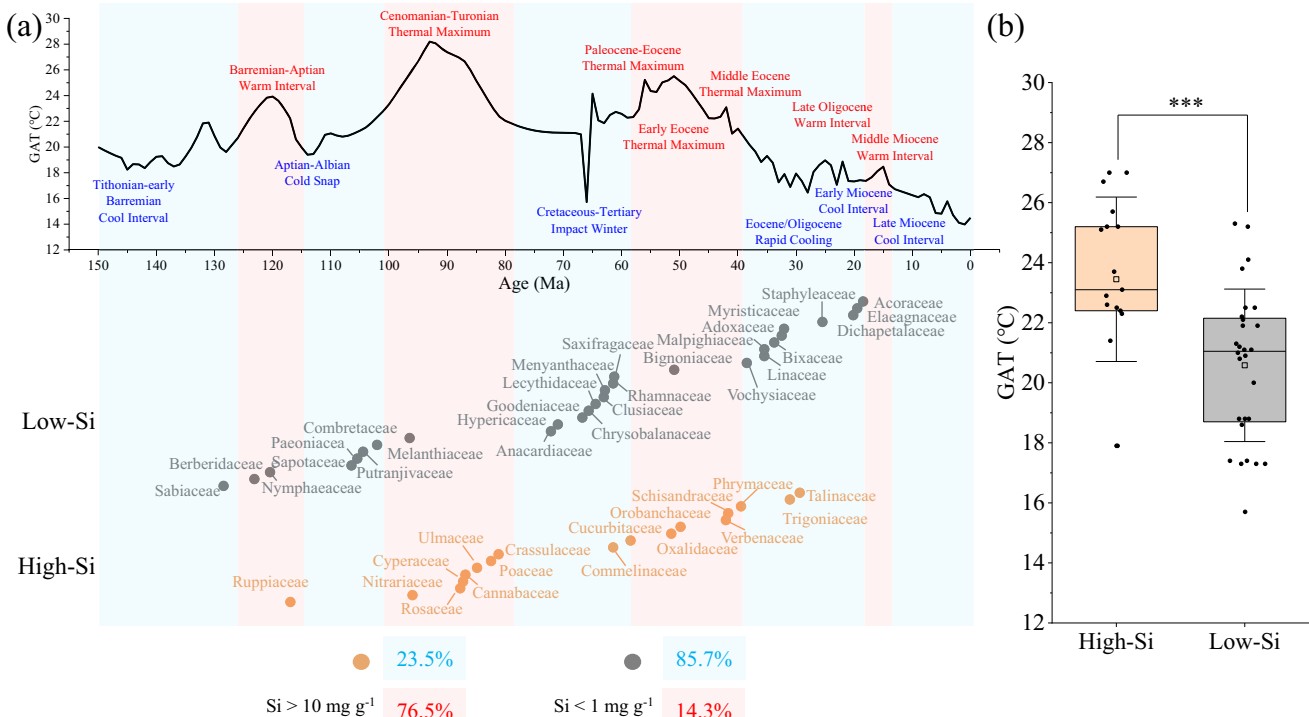

**Fig. 4 | Evolution of high-Si and low-Si angiosperm families in relation to temperature fluctuations through Earth history. a** Emergence of plant families with different leaf Si concentrations during warming and cooling episodes. Evolutionary information was obtained from Angiosperm Phylogeny Website (www.mobot.org/MOBOT/research/APweb/). Each point represents a family. To avoid point overlap, there is a one unit offset between different points on the y-axis of the lower part of figure. Families with leaf Si concentration above 10 mg g⁻¹ are marked orange ($n = 17$), and families with leaf Si concentration below 1 mg g⁻¹ are marked gray ($n = 28$). GAT means the global average temperature in history. Red areas indicate warming episodes and blue areas indicate cooling episodes, and the division is based on Scotese[36,37]. **b** Earth's temperatures during the emerging time of high- ($n = 14$) and low-Si ($n = 28$) families. In box plots, the central box represents the interquartile range, bounded by the first (25th percentile) and third (75th percentile) quartiles. The horizontal line inside the box indicates the median (50th percentile). The hollow rectangle and whiskers represent the mean and standard deviation of the data. Differences between high- and low-Si families are compared using Welch two-sided t-tests ($p = 0.0009$). Significant differences are indicated as follows: \*\*\*$p < 0.001$.

We used a recently published Si database[12] containing leaf Si concentrations of 1826 species and 213 families to analyze the relationship between the evolution of high-Si (>10 mg g⁻¹) and low-Si (<1 mg g⁻¹) angiosperm families and temperatures through Earth history (Supplementary Fig. S6, see Supplementary Data 2 for leaf Si concentrations). During Earth history, climate has repeatedly alternated between warming and cooling episodes[36]. In this paper, this division is redrawn based on Scotese et al.'s paleotemperature timescale figures and the original data[36,37] (Supplementary Fig. S7, see Supplementary Data 3 for Earth history temperatures). Using data from the Angiosperm Phylogeny Website, we investigated the occurrence of plant families with contrasted Si concentrations during Earth warming and cooling episodes. We found that, among families for which phylogenetic time for crown-group radiation can be obtained, 77% of high-Si families ($n = 17$) emerged during warming episodes, while 86% of low-Si families ($n = 28$) emerged during cooling episodes (Fig. 4a). The historical temperature of the Earth during the emergence of high-Si families was 3 °C higher than that of low-Si families ($p < 0.001$) (Fig. 4b, see Supplementary Table S4 for specific information on the plant families).

As there is significant uncertainty in origin of angiosperms[38], we selected another set of phylogenetic time data from Li et al. for a second analysis[39]. Unlike the evolutionary information collected from different literature sources on the Angiosperm Phylogeny Website, the data from Li et al. was from their plastid phylogenomic tree, which was based on 80 genes from 2881 plastid genomes and 62 fossil calibrations. A similar result was obtained, where high-Si families were more likely to appear during warming episodes, while low-Si families mainly appeared during cooling episodes (Supplementary Fig. S8).

Within each clade, there are often significant differences in Si concentrations among different sub-clades (e.g., subfamilies within a family). To further test the hypothesis that temperature drives the evolution of plant Si, we selected five angiosperm families (including Asteraceae, Orchidaceae, Fabaceae, Rubiaceae, and Poaceae) to separately analyze the emergence time of their different sub-clades. They are the most species-rich families within the angiosperms, with over 10,000 species each[40,41]. The detailed information on the leaf Si concentrations and differentiation time of the lower taxonomic units of the five major angiosperm families is shown in Supplementary Table S5.

In Asteraceae, species with Si concentrations higher than 10 mg g⁻¹ are found in the Asteroideae subfamily, which emerged during the Middle Eocene Thermal Maximum[42,43], while species with Si concentrations lower than 2 mg g⁻¹ are typical of the Carduoideae and Cichorioideae subfamilies, which emerged in cooling episodes[44,45]. In Orchidaceae, Si-rich species are found in the Apostasioideae and Cypripedoidoidae subfamilies, which diversified during a warming episode, whereas Si is notably absent in the Vanilloideae, Orchidoideae, and Epidendrioideae subfamilies that emerged during a cooling episode[46,47]. Fabaceae exhibits low Si concentrations (<1 mg g⁻¹) in subfamilies like Caesalpinioideae and Cercidoideae that appeared during cooling episodes, while high-Si species are mainly in the Papilionoideae, particularly the Phaseoleae subclade that emerged during a warming episode[48,49]. In Rubiaceae, high-Si species are found in the Rubioideae, which emerged in a warming episode, while low-Si species appear in the Ixoroideae and Cinchonoideae, which emerged during a cooling episode[50]. Among Poaceae, low-Si species distributed among five sub-clades (Poodinae, Panicoideae, Poeae, Chloridoideae, and

Paspaleae) all emerged during cooling episodes, except Chloridoideae. By contrast, high-Si species mainly belong to the Arundinarieae, Panicoideae, Cynodonteae, and Molinieae, of which all but the Cynodonteae emerged during warming episodes.

Overall, our analysis of the sub-clades from five different major families of angiosperms showed that the leaf Si concentrations in sub-clades seem to be closely related to the Earth's temperature during their appearance period. This suggests that silicification might have been favored during warm and interglacial periods and/or not favored during glacial periods.

## Discussion

Our research suggests that temperature plays a significant role in plant Si accumulation. We first highlighted the beneficial effects of Si for plant growth at high temperatures and its harmful effects at low temperatures in an emblematic Si-accumulating species, rice, with consistent positive effects across both lowland and upland varieties. We also demonstrated that increases in temperature promote Si accumulation in rice, under controlled humidity. We then found that the average temperature of the present-day distribution of high-Si plant clades is higher than that of low-Si ones, and that the intraspecific Si variation is positively correlated with temperature for two high-Si species. Following that, we traced back through history and found that both the evolution of Si transporters and the differentiation of high-Si species were related to the warm periods of the Earth.

Silicon can help plants tolerate high temperature stress in a range of environmental conditions, whether rice grown in paddy fields[51,52] or crops grown in dryland, such as wheat[53], barley[54], tomatoes[55] and soybeans[56]. Numerous processes have been suggested to explain this beneficial effect[20,22]. The presence of silicified structures in leaf epidermis could significantly reduce heat load and prevent ultraviolet radiation by effective emission of thermal energy[6,21,57]. In addition, Si in rice leaves can prevent electrolyte leakage and maintain leaf water content by providing mechanical support to cell walls[52]. Silicon could also restore ultrastructural distortions of cellular organelles provoked by heat stress, particularly chloroplasts and the nucleus, although the mechanism is not yet fully understood[53]. Moreover, the enhancement of plant antioxidant defense system in Si-treated plants has been reported to resist oxidative damage caused by high temperature[58,59]. Although our understanding of the role of Si in regulating heat stress remains incomplete, these mechanisms suggest that high temperatures might have favored the emergence of silicification.

Silicon can help plants resist high-temperature stress, but could have an opposite effect on cold and freezing stress. Preserving cell integrity is critical for plants under freezing stress[60], but the thermal expansion of silica deposited in cell walls is much lower than that of cellulose and lignin, and therefore may damage the cell structure due to deformation differences caused by the effects of low temperature[61,62]. In addition, intracellular ice formation poses a greater threat to plants than extracellular ice formation, and plants use a mechanism called deep supercooling to prevent it, enabling them to tolerate low temperatures[63]. However, Si deposits in cell walls inhibit cellular water loss[64], which could interfere with deep supercooling and increase the risk of intracellular ice formation. In addition, ice formation in plants is a dynamic process: water remains in a metastable liquid state until it turns into ice around an ice-nucleating substance, and then propagates longitudinally and radially[65]. Silicon has also been proven to act as an ice-nucleating substance to accelerate freezing[66], which may be one of the reasons why ice propagation is found to be more rapid in Poaceae (high-Si) than in Gymnosperms and Dicotyledons (low-Si)[67,68].

Temperature affects plant Si uptake by influencing energy consuming Si transporters such as Lsi2[15,16], or by affecting stomatal conductance and transpiration[69]. However, stomatal conductance and transpiration are also affected by humidity[70], so when analyzing the

mechanism by which temperature affects plant Si uptake, it is usually confounded by effects of humidity. We have tested that possibility through the inclusion of nine climatic variables into the multiple regression analysis and found that MAT is the most important driver of Si accumulation. In addition, our experimental results indicate that, under the same humidity, an increase in temperature can also promote the accumulation of Si in rice, but not in other mineral elements. Therefore, in our study, the relationship between high temperature and plant Si accumulation can reasonably exclude the influence of humidity.

We also considered the relationship between the evolution of plant silicification and paleoclimatic changes throughout history. First, we should note that leaf Si concentrations are closely linked to the Si uptake mechanisms[16]. The evolution of NIP-IIIs, serving as primary Si influx transporters, traces back to approximately 515 million years ago (Ma), during the Middle-late Cambrian period, characterized by a hot climate[71]. The evolutionary trees we constructed for the currently reported Si transporter proteins (Lsi1, 2, 3, and 6) collectively show that Si transporter proteins in high-Si taxa within the Poaceae evolved mainly during warm climatic periods. However, another hypothesis is that these proteins and the genes coding for them might have evolved before high-Si plants evolved, perhaps because they all have to be in place before high Si deposition is possible. Therefore, more fossil evidence and research involving more species beyond the Poaceae are needed to understand the factors driving the evolution of Si transporters more fully.

Throughout the algae and plant kingdom, it is often observed that high-Si clades emerge during the Earth warming episodes, while low-Si clades emerge during cooling episodes. For example, among algae, the marine planktonic algae (Archaeplastida), which are low in Si, emerged between 659 and 645 Ma ago, during the Snowball Earth period[72]. However, diatoms, which are very rich in Si, originated within this group 200 Ma ago during the Triassic Hothouse[73]. In addition, both mosses and ferns exhibit high leaf Si concentrations (Supplementary Fig. S6), and recent dating of the land plant phylogeny suggests that these clades may have emerged as early as during Cambrian–Ordovician Hothouse[74]. A transition from fern-dominated ecosystems (high Si) to gymnosperms-dominated landscapes (low Si) coincided with the Earth's shift from the Late Devonian Hothouse to the Late Paleozoic Icehouse[75]. Subsequently, Angiosperms rapidly diversified, dominating terrestrial ecosystems since the Lower Cretaceous period[39]. The emergence of early angiosperm taxa, including Amborellales, Nymphaeales (low-Si), and Austrobaileyales (high-Si), has also been linked to climate, with Nymphaeales appearing in cooler intervals and Austrobaileyales during warmer times. Further, our analysis, based on two different phylogenetic databases of angiosperms, also supports the trend of high-Si families evolving in warming episodes and low-Si families in cooling episodes (Fig. 4b, Supplementary Fig. S8).

Taken together, these results suggest that plant silicification is likely to be influenced by paleoclimatic changes. From a long-term evolutionary perspective, the different effects of Si on plant performance at high and low temperatures could provide an explanation for plants being selected to accumulate Si or not. However, we should be aware that there may be other potential factors affecting plant silicification, such as soil fertility, humidity (or drought), herbivores, habitat types, photosynthetic pathways and $CO_2$ levels[12,18,76-79]. These published studies have led to several different hypotheses regarding the evolution of high Si levels in plants, but many remain untested. Furthermore, most current evidence supports phenotypic plasticity in response to these factors rather than that they are drivers of natural selection for silicification[11,19].

One hypothesis suggests that silicification might have evolved during arid periods[80,81], because Si can enhance drought resistance in rice[82,83], as well as in sorghum[84] and sugarcane[85], and some studies have found that leaf Si concentration is higher under drought conditions[86,87]. However, many laboratory and field experiments have shown that

drought reduces the leaf Si concentration[88,89], whilst the Si concentration of upland rice, grown in rain-fed conditions is often lower than that of lowland rice, usually grown submerged[90–93]. Overall, these studies could reflect phenotypic plasticity and the influence of drought on soil Si availability and/or plant transpiration, and more research is needed to decipher the interrelationships between drought and temperature in affecting plant silicification.

Another hypothesis suggests that selection for high Si taxa is related to herbivory: high levels of herbivory lead to increased Si uptake to deter feeding[94]. Although studies have found that damage to plants by rodents and insects increases the absorption and accumulation of Si in grass[95,96], large mammalian herbivores[77,97,98] or artificial clipping[99] do not increase the absorption and accumulation of Si to the same extent[100]. These lines of evidence show that Si may be an effective anti-herbivore defense in the present day, at least against some types of herbivore, but evidence that herbivory drove the adaptive origin of this trait is less convincing. For example, phylogenetic and fossil evidence fail to show a close correlation between grass-feeding herbivores, grass dominance and phytolith content in grasses: open habitat grasses that would be expected to have co-evolved with grazing large herbivores do not have higher Si content, rejecting the idea that such herbivory promoted natural selection for higher Si levels[11].

At the same time as the adaptation of ungulate herbivores to C4-grass dominated habitats, some lineages of C4 grasses with particularly high levels of phytolith deposition emerged, leading to speculation that C4 grasses have experienced stronger selection for Si-based defenses than C3 ones[101]. However, other studies have not found consistent differences in Si concentration between C3 grasses and C4 grasses[11,19]. Furthermore, C4 grasses, particularly the lineages with high levels of phytoliths, are typical of hot, dry habitats, so climatic factors, including temperature, could have been the driver of high Si levels in these species. Other studies also support this idea and have found that high temperature is associated with increased Si concentration, especially in taxonomic groups adapted to warm regions[19], although there are also cases where the effect of temperature on Si levels among grass species is less consistent[24].

Another potential evolutionary driver of increase in silicification is atmospheric $CO_2$. During periods of low $CO_2$ levels in the atmosphere, plants would use Si to replace C, which is more energy-efficient[12,18]. Increased $CO_2$ levels have been shown to reduce Si concentrations in plants, but again evidence that this phenotypic response drives selection is lacking: there is no correlation between evolution of active Si-deposition in vascular plant branches and the drop in atmospheric $CO_2$ during the Carboniferous period, indicating that the reduction in carbon acquisition pathways did not lead to the widespread use of phytoliths in land plants[11]. In addition, $CO_2$ levels tend to be correlated with high temperatures (although not always perfectly)[102], which might suggest an alternative, temperature-based mechanism for the correlation between high $CO_2$ and the emergence of high-Si families.

Overall, while there is evidence that other factors influence plant Si concentration through plasticity, our analysis provides the strongest evidence to date that global temperatures offer an evolutionary explanation for the emergence of plant silicification. The Si cycle is crucial for regulating the Earth's C cycle[103]. Silicate weathering fixes $CO_2$ from the atmosphere, and plants enhance this process through various mechanisms[104] that are increasingly implemented in agroecosystems worldwide for both carbon sequestration and crop yields[105–107]. Our findings on the link between plant silicification and changes in the Earth's temperature provide new clues regarding the interactions between Si and C cycling over geological timescales[13].

## Methods

### Effect of Si on rice's response to high and low temperature stress

We selected rice (*Oryza sativa* L. spp. *japonica*, var Nipponbare), a typical high-Si species, for the experiment testing the effect of Si on high and low temperature stress. To kill the microorganisms on the surface of the seeds to prevent them from becoming mouldy during germination[108], seeds were soaked in 10% $H_2O_2$ (v/v) for 10 min, and then rinsed with deionized water five times[109]. They were placed on moist filter paper and germinated at 30 °C in the dark for three days. Seeds were then transferred to black plastic pots containing 1.2 L of deionized water for five days. Next, seedlings were grown in 1/4-strength Kimura B solution (adjusted to pH 5.6), renewed every five days in a growth chamber (at 30 °C, 55% relative humidity, and using a 14/10 h light/dark period). In order to have a sufficient number of healthy seedlings, we prepared 30 pots and planted 10 seedlings in each pot. After 20 days, 120 uniform-sized seedlings were selected for the experiment.

Rice seedlings were divided into two groups: the +Si group, for which 1.0 mM of silicic acid (produced by passing sodium silicic acid solution through a column packed with ion exchange resin) was added to the nutrient solution, and the -Si group, which did not receive silicic acid. Silicic acid spontaneously aggregates at concentrations above 2.0 mM[110], and 1.0 mM has been proven to be the most beneficial to plant growth in previous rice experiments[111]. Apart from Si inputs, growth conditions were similar. The nutrient solution was renewed every five days. After 15 days, there was no difference in shoot length, root length, fresh weight, and chlorophyll concentration between the two groups (Supplementary Table S1). We then changed the nutrient solution for both the +Si and -Si groups with 1/4-strength Kimura B nutrient solution without Si, and placed them in constant temperature incubators under high temperature (40 °C), low temperature (0 °C), or freeze thawing (−2 °C for 12 h and 2 °C for 12 h each day) stress. Each treatment included 5 pots, with 4 plants planted in each pot. Three days later, we took photos for rice and measured the shoot length (the length of the highest shoot), root length (the length of the longest root), fresh weight (the sum of shoots and roots), and chlorophyll concentration (using SPAD-502 chlorophyll meter, chlorophyll concentration can be obtained from the following formula: Chlorophyll concentration (mg g$^{-1}$ fresh weight) = 0.965 exp (0.036 SPAD value) − 1)[112]. We randomly selected one plant from each pot for measurement, with a total of five replicates for each treatment.

In addition, we also conducted similar studies on rice varieties using lowland rice, *indica*, cv. Jinzao 47, and three varieties of upland rice: *japonica*, cv. Linhan 1, *japonica*, cv. Danhan 53 and *japonica*, cv. Zhenghan 10. We cultivated 10-day old rice in nutrient solution with/without 1.0 mM silicic acid for 15 days before stress. Then, the rice was subjected to high temperature (40 °C) and low temperature (0 °C) stress. Three days later, we took photos and measured shoot length, root length, fresh weight, and chlorophyll concentration. Seed germination, growth conditions, and replicate settings for each variety of rice were all consistent with those of Nipponbare. Data were expressed as mean ± standard deviation of five replicates. Two-sided Welch t-tests were used to test for significant differences between groups for each stress treatment. The bar charts were drawn using Origin 2021.

### Effect of temperature on silicon accumulation in rice

We chose rice (*Oryza sativa* L. spp. *japonica*, var Nipponbare) as the experimental material to study the effect of temperature on silicon concentration in rice leaves. The conditions for seed germination and seedling growth were the same as before. We prepared 12 pots, each pot planting four 20-day old rice seedlings with similar growth conditions. Due to the optimal growth temperature range of rice being 25–30 °C[27], in order to avoid growth inhibition caused by temperature stress, we chose 26 °C and 28 °C for the experiment. Six pots of rice were placed in a constant temperature incubator at 26 °C, and the other 6 pots in a constant temperature incubator at 28 °C (using humidifiers to control the humidity to 70%). Each pot contained 1 L of 1/4-strength Kimura B solution (adjusted to pH 5.6) with 1.0 mM silicic acid, and the nutrient solution was replaced every 5 days. After 10 days

of cultivation, rice leaves were collected and dried. Then, we weighed about 0.1 g of dry leaf samples and recorded their specific weights. The leaves were dissolved with 4 ml of $HNO_3$, 2 ml of $H_2O_2$, and 0.5 ml of HF (for specific dissolution of Si)[27]. After diluting the solution to 10 ml with 5% volume fraction boric acid to eliminate residual HF[113], we used an inductively coupled plasma optical emission spectrometer (ICP-OES; PE-7000DV, USA) to measure the concentrations of elements such as silicon, phosphorus, potassium, calcium, magnesium, manganese, copper, and zinc in the digestion solution, and calculated the elemental concentrations in the rice leaves. We selected one plant from each pot for measurement, and each treatment had a total of six replicates. The data was represented as mean ± standard deviation. Two-sided Welch t-tests were used to test for significant differences in element concentrations in rice leaves under different temperature treatments.

## Distribution temperature of high- and low-Si plants
Based on Hodson's analysis of different clades' leaf Si concentrations[8], we selected the top 10 Si concentration ranked clades as the orders with relatively high Si concentration (abbreviation as high-Si clades) and the bottom 10 Si concentration ranked clades as the orders with relatively low Si concentration (abbreviation as low-Si clades) for subsequent analysis. The high-Si clades include: Poales, Saxifragales, Schisandraceae, Arecales, Boraginaceae, Fagales, Rosales, Nymphaeaceae, Asterales, Alismatales. Low-Si clades include: Acorales, Brassicales, Liliales, Asparagales, Aquifoliales, Santalales, Pandanales, Celastraceae, Cornales, Bromeliaceae. We extracted the location of 1000 occurrences randomly selected within each clade from the Global Biodiversity Information Facility, using the rgbif package in R (https://www.gbif.org/developer/occurrence#search, see Supplementary Code 1 and Source Data for details). After data cleaning, the final 19962 occurrences were used for subsequent analyses. We obtained mean annual air temperatures (MAT) from Climatologies at High Resolution for the Earth's Land Surface Areas (CHELSA, https://chelsa-climate.org/) to study the temperature distribution of high- and low-Si plants[114]. Then, the average, median, and mode of distribution temperature of high- and low-Si plants were tested. Finally, we mapped the world and compared the distribution temperature of high- and low-Si plants by two-sided Wilcoxon rank sum test using R 4.3.1 (Supplementary Code 1)[115].

## Effect of climate variables on phytolith/Si concentrations
To consider the intraspecific variation in leaf phytolith/Si concentrations, a sampling campaign was conducted in China on two typical high-Si plants (wheat and rice) and two low-Si plants (weeping willow and winter jasmine). We collected 475 wheat individuals, 279 rice individuals, 140 weeping willow individuals, and 80 winter jasmine individuals across China. The distribution of the sampling sites is 86.02-126.72 °E and 19.49-45.74 °N (Fig. 3a). The mapping of China was done using R 4.3.1. The Chinese map data is sourced from DataV.GeoAtlas (http://datav.aliyun.com/portal/school/atlas/area_selector). Leaves were dried (105 °C for 30 min, and then 75 °C for 48 h) and ground through a 100-mesh (0.15 mm) sieve. Research has shown that phytoliths are the main form of silicon deposited in higher plants, accounting for more than 90% of total Si in plants[116]. Therefore, there is a significant correlation between the Si concentrations and the phytolith concentrations of crop leaves[117–119]. Leaf phytoliths were extracted with $HNO_3$, HCl, and $H_2O_2$ by microwave digestion method[120,121]. Extracted phytoliths were washed with deionized water and dried in an oven at 75 °C for 48 h, and weighed. In the case of plants such as weeping willow and winter jasmine, the low Si content makes it impossible to extract phytoliths from their leaves, so total Si concentration was determined as follows. The leaves were digested in the Microwave Digestion System using a mixture of 4 ml of $HNO_3$, 2 ml of $H_2O_2$, and 0.5 ml of HF[27]. After reacting HF with 5% volume fraction boric acid and making a constant volume of 10 ml,

the Si concentration was determined using the ICP-OES (PE-7000DV, USA).

Then, we obtained climate variables of sampling points from Resource and Environmental Science Data Center (RESDC, http://www.resdc.cn/DOI) and CHELSA, including mean annual air temperature (MAT, °C), ground surface temperature (GST, °C), evaporation (EVP, mm), precipitation (PRE, mm), pressure (PRS, hPa), relative humidity (RHU, %), wind speed (WIN, m/s), surface downwelling shortwave flux in air (RSDS, $MJ\ m^{-2}\ d^{-1}$), and vapor pressure deficit (VPD, Pa). We first analyzed the correlation between MAT and leaf phytolith/Si concentration using R 4.3.1. Then, multiple regression models were implemented using ordinary least squares (OLS) in R 4.3.1 (Supplementary Table S2, Supplementary Code 2)[29]. All variables were standardized to a mean of 0 and SD of 1 before conducting the regression analysis. In addition, we also assessed the relative importance of various climate variables in explaining the phytolith concentration in wheat and rice leaves by using the squared standardized coefficients (betasq) method of relaimpo package in R 4.3.1 (Supplementary Code 2)[122].

## Differentiation of Si uptake and transport proteins and historic climate change
To investigate the evolution of Si transporters in higher plants, we focused on four key proteins responsible for Si absorption and transport (Lsi1, Lsi2, Lsi3, Lsi6). To construct a timetree of each one of four Lsi proteins, 20 protein sequences of different species with highest similarity to that of rice (*Oryza sativa* ssp. *japonica*) were obtained through BLASTP (blast.ncbi.nlm.nih.gov)[123], with homologous proteins of *Arabidopsis thaliana* as outgroups (see Supplementary Data 1 for the sequence information). A Maximum-Likelihood phylogenetic tree was constructed using MEGA11[31] with most suitable amino acid substitution model selected via ProtTest (version 3.4.2)[32]. The corresponding timetree was constructed via Phylogenetic Analysis by Maximum Likelihood (PAML)[33], with differentiation time of Lsi proteins estimated by MCMCtree[34], a program applying the Bayesian Markov Chain Monte Carlo (MCMC) method, and checked by pairwise divergence time of 3 couples of species provided by Timetree (https://timetree.org/)[35]. The pairwise divergence time used to construct the timetree for each protein is detailed in Supplementary Table S3. Note that the outgroups of each tree were omitted to highlight the species of interest, as well as better fit the Earth's paleotemperature graph (Supplementary Fig. S5).

We used the dataset of historic climate change over the last 200 Ma reported by Scotese et al.[36,37] (Supplementary Data 3), which distinguished between Earth warming and cooling episodes based on a combination of lithologic evidence and oxygen isotope methods. We directly drew on the classification of warming and cooling episodes in literature and matched the historic climate with the evolutionary trees to analyze the relationship between protein differentiation and temperature.

## Differentiation time of high-Si and low-Si plants and historic climate change
We obtained information on leaf Si concentrations in different species and families from de Tombeur et al.[12] – the largest database on leaf Si concentrations ever published. Taxonomic information was corrected using the NCBI database (www.ncbi.nlm.nih.gov). For each plant family within the pteridophytes, gymnosperms and angiosperms, we calculated the mean and standard errors of leaf Si concentrations (Supplementary Data 2). All analyses were performed in R 4.3.1[115] (Supplementary Code 3).

To test the link between the evolution of Si accumulation in plants and historic temperature fluctuations, we obtained evolutionary time data for both high- ($>10\ mg\ g^{-1}$) and low-Si ($<1\ mg\ g^{-1}$) families from Angiosperm Phylogeny Website (www.mobot.org/MOBOT/research/APweb/) (Supplementary Table S4). We then

compared Earth's historical temperatures during the emergence of high- and low-Si families. The box plot was drawn using Origin 2021. Given that the exact timing of species divergence may differ in different databases, we conducted the same analysis using another phylogenetic database from Li et al.[39], and obtained similar results (Supplementary Fig. S8).

Within the five families with the largest number of species among the angiosperms (Asteraceae, Orchidaceae, Fabaceae, Rubiaceae, and Poaceae), there was a large variation in leaf Si concentrations among the different sub-clades[12]. We first collected the divergence times of different sub-clades by searching for keywords of the five major families in the web of science database (www.webofscience.com/). We then analyzed the relationship between the mean leaf Si concentration of different sub-clades, their divergence dates and historic climate[36] (Supplementary Table S5).

### Reporting summary
Further information on research design is available in the Nature Portfolio Reporting Summary linked to this article.

## Data availability
The accession numbers of Si transporters on NCBI are: NP_001403822.1 (Lsi1), NP_001388947.1 (Lsi2), BAS18932.1 (Lsi3), NP_001389719.1 (Lsi6). All data generated in this study are provided in the Supplementary Information/Source Data file. Source data are provided as a Source Data file Source data are provided with this paper.

## Code availability
All code generated in this study are provided in the Supplementary Information/Source Data file.

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

## Acknowledgements

This work was jointly supported by grants from National Natural Science Foundation of China (324B2064, Z.P. and 32272799, Y.L.), the Fundamental Research Funds for the Central Universities (226-2024-00052, Y.L.), National Key Research and Development Program of China (2023YFD1900601, Y.L.), and the SNSF Ambizione Fellowship Program (PZ00P3_193646, C.M.Z.). In addition, we appreciate the contribution of Lixue Qiu, Li Tan, Wenjuan Li, Enqiang Zhao, Alin Song, Ning Ling, Wei Liu, Hong Pan, Tao Liu, Ling Xiao, Jianping Yang, Jianxiao Wang, Zhuoxi Xiao, Weifeng Xu, Zhenhua Zhang, Gaofei Ge, Ying Zhang, Ping Li, Qi Tao, Xiaozhong Wang, Mujun Ye, Yiyong Zhu, Xiaoling Yang, Peiyuan Cui, Jie Zhang, Zhiyang Jiang, Zhaohui Wang, Jinjia Gan, Jiawei Shi, Liqin Gao, and Zhizhen Ye, during the sampling process.

## Author contributions

Y.L. and N.N. planned and designed the study with equal contributions and wrote the paper. Z.P., F.d.T., and S.E.H. wrote the paper. Z.P., Y.W., and P.Z. performed data analysis. C.M.Z., M.N., C.V., L.M., T.W.C., D.X.G., Z.L., Y.G.Z., H.P., and C.A.E.S. performed review and editing.

## Competing interests

The authors declare no competing interests.
