## [Transparent Peer Review file · Nature Communications]

Convergent evidence for the temperature-dependent emergence of silicification in terrestrial plants

Corresponding Author: Professor Yongchao Liang

Version 0:

Reviewer comments:

Reviewer #1

(Remarks to the Author)

The manuscript submitted to Nature Communications represents valuable contribution discussing and critically reviewing the knowledge of Si accumulating plants in relation with temperature. The authors analyse the distribution of these species from the viewpoint of systematic botany and historical phylogenetical development of plant species. The results are surprising and future discussion about this topic can be expected. The authors add results of one experiment in addition to the review of present knowledge.

The text is generally well written and easy to follow. I have just a few comments and suggestions to add some data for extending the problem more broadly in relation with drought often accompanying climate warm periods and may be some additional data about Si accumulating species can be mentioned.

The authors do not mention several publications dealing with the role of Si cycle regulating the Earth's carbon cycle e.g.: Hodson, M.J., Guppy, C.N. Some thoughts on silicon and carbon trade-offs in plants. *Plant Soil* 477, 233–239 (2022). <https://doi.org/10.1007/s11104-022-05394-5>

Some additional data about experiments confirming the role of Si for sorghum and sugarcane exposed to drought can be mentioned or discussed in the chapter Discussion.

Rice (*Oryza sativa* L. cv. Nipponbare) was selected for the experiment testing the effect of Si on high and low temperature stress. This cultivar, often used for experiments is a typical lowland cultivar. Comparison (of Si accumulation or even growth reaction) with some upland cultivar adapted to different water and temperature conditions may bring some new aspect to the present contribution. The data about the difference of Si accumulation in lowland and upland rice cultivars are known from the literature.

Specific comments

Fig. 3 MAT - better write: mean annual temperature – it will be more clear for the reader.

Reviewer #2

(Remarks to the Author)

**** Major Comments ****

This manuscript addresses the evolutionary origins of plant silicification, an important, timely, and moderately controversial topic in plant evolution and ecology. There is a lot to like about this manuscript and I appreciate the multiple sources of data that were used to address the question of why plants have evolved to concentrate silica in their tissues. There are literally hundreds of studies which have quantified silica concentration in plants and dozens which have proposed ecological and evolutionary drivers of plant silicification. Special topic papers or reviews in high profile journals over the last ten to fifteen years highlight the lack of consensus on why some plants concentrate Si in their tissues and others do not. Three issues

have traditionally plagued these studies: (i) the various proposed drivers of plant silicification are often correlated and difficult to tease apart, (ii) attempts to answer this question have been largely observational and have lacked experimental rigor, and (iii) the hypotheses underlying these explanations have lacked a clear mechanistic evolutionary explanation (i.e., an ultimate rather than proximate cause).

This paper attempts to overcome points (ii) and (iii), with less of an attempt to overcome point (i).

Past studies that have failed to address point (i) have focused on one mechanism and have sought to provide evidence for that single mechanism, rather than trying to eliminate alternative competing mechanisms that may be correlated but not ultimately responsible for the observed increase in Si. Right now, there is a long list of potential reasons Si accumulation may have evolved in plants. This remains an important setback for this paper. In fact, parts of their analysis, which I address below, seem to cover up the fact that in observational studies other important environmental drivers, such as soil or potentially rainfall, contribute to observed phytolith concentrations in crop systems. To me a goal of this study, and worthy of publication in a distinguished journal like *Nature Communications*, should be to eliminate alternative mechanisms in favor of temperature being the principle and ultimate evolutionary factor responsible for plant silica concentrations in deep time. To do this will take some work because it will mean bringing other, potentially correlated, variables into the analysis and asking which is the best explanatory variable. If you cannot eliminate those other potential variables, then all you are doing is throwing one more variable onto an already large pile of possible explanations. I have three major suggestions for this manuscript:

(1) My first major suggestion for this paper is to bring other variables that can account for variation in plant silicification into the observational regression analysis presented in Figures 3 and S3. Typically, I try to review papers based on the work done, rather than on work that was not done. However, in this case it is essential to the nature of the argument that other factors be discussed and possibly eliminated as explanations for plant silicification. Consequently, other environmental factors need to be included in the observational analysis of phytolith concentration across China and a model selection procedure should be used to identify the best explanation for the response. Second, across the MAT gradient in China I urge the authors to find a parallel dataset for no-Si accumulating crop species. Implicit in their hypothesis that evolutionary periods of high temperature are responsible for Si-accumulation in plants is that plants which evolved during cool periods should not respond to gradients of temperature by altering Si concentrations in their tissue. This can and should be explicitly tested by including an analysis of non-Si accumulating plants across the same gradient of MAT.

(2) My second major comment is to provide better justification for some of the decision points in the manuscript which can be constructed as arbitrary. These include (i) the thresholds of high (>2 mg g⁻¹) and low (<1 mg g⁻¹) silica concentrations, (ii) taxonomic analysis (i.e., which plant families or subfamilies were included in the analysis), and (iii) the temperature cutoffs for warm and cool periods (18o C and which lineages arose in warm or cool periods).

(3) My third major comment is to provide better descriptions of the methods, statistical analyses, and results. I urge the authors to add much more information and detail into future versions of this manuscript. Telling readers "... rice plants were placed under three types of stress (heat, cold, and freeze thawing) for three days" is not detailed enough for an outlet like this. Papers I have read and written for these condensed format journals are packed with information, and for good reason – space is limited. But as it currently stands there is nowhere near enough information provided for readers to understand the analysis or judge the validity of what was done. The lack of information about basic methods and results, which I comment on in detail in the *Specific Comments* section, is not at a level that is acceptable for a peer-reviewed outlet. I understand well the "iceberg" analogy for papers formatted for science and nature, but that does not mean it is acceptable to leave essential information out of the "tip of the iceberg" which is published in the main journal. To continue this analogy, the submerged part of the iceberg (i.e., the supplemental materials) in this manuscript is a disappointing fourteen pages including several long tables filled with information about various plant families. The content presented in the methods and supplemental materials are insufficient and do not currently represent reproducible research.

There are several specific issues with the presentation of results, data, and statistics. Most (but not all) of these issues can likely be dealt with by providing more information. The manuscript lacks basic reporting of statistical tests and associated test statistics and the assumptions (or violations) of statistical tests that were used. My suggestion is that any time you report results tell your reader what analysis was used to arrive at that result along with any associated test statistics or diagnostic information. Then refer the reader to the methods or supplementary materials where you can provide all the details of the test and exactly how it was carried out – to the point that the reader could replicate exactly what was done if they so choose. The paper needs to be re-written to include all the methodological details of the paper in expanded form, either in the methods or in supplemental materials. Moreover, I urge the authors to consider including data and scripts so that all analyses are reproducible. Publishing fully reproducible research with code and data to rerun all the analyses is the gold standard for scientific publishing for a journal such as *Nature Communications*. This manuscript has a long way to go.

** Specific Comments **

Abstract: define "high temperature stress", "warm geological periods" and "cold geological periods" in the abstract. These relative terms really need to have values associated with them to be meaningful.

Lines 55 – 56: the treatments need to be stated explicitly here.

Lines 60 – 61: is the intention to relate the freeze thawing to chlorophyll concentration? If so, you need to tell your reader explicitly what you want them to know – don't make them connect the dots on their own.

Line 62 & Fig. 1: in panel (d) should the label be related to Si concentration? No one is going to know what SPAD means here. It should be replaced with something meaningful.

Lines 64 - 67 & Fig. 1 legend: even though the temperature treatments are located (inconspicuously) in the figure the specific treatments should be clearly indicated here.

Line 74: this is what I mean about the lack of details in this manuscript. Tell your reader the exact number of species.

Lines 78 – 83: how did you arrive at these orders? What was the procedure for selecting these orders for analysis and what justification can you provide for the focus on these taxonomic groupings.

Line 88 & Fig. 2b: is the average the correct metric by which to compare these distributions? What statistical test did you use for this comparison – again, the authors have not provided sufficient information to understand what was done or to judge the validity of the analysis. The legend only tells us that there is some statistical difference at $p < 0.001$ with no information anywhere on the test, test statistics, degrees of freedom, or assumptions of the test (i.e., normality). This is unacceptable for a journal of this stature. The sample sizes in this analysis are enormous (~10,000!), meaning if they did use a parametric statistical test like a t-test then very small differences in the means would be deemed statistically significant. As a quick demonstration: two randomly generated normal distributions each with $n=10,000$ and means of 14.9 and 15.1 (assuming a standard deviation of 5, thereby producing a range similar to your temperature values) will be statistically significantly different at $p < 0.001$ approximately 32% of the time. I don't think many biologists would argue that mean temperature values of 14.9 and 15.1 are meaningful differences! And this demonstration was done with normally distributed errors. The error distribution of your data appears not to be normally distributed, making the error even worse. The information we need is about the values of the central tendency and what is the effect size of their difference. Or in this case perhaps it is more appropriate to compare distributions with a non-parametric test or analyze modes, which appear at different temperatures for High-Si and Low-Si.

Lines 93 – 97: I said in my major comments that I do not like to evaluate a manuscript based on what was not done, but in this case, it seems pertinent because (i) the many other potential mechanisms which have been proposed in the past and (ii) the bold claims this paper is making about temperature as the major evolutionary force. Most ecologists interested in this subject would love to see a really strong and definitive case for an answer (me included), but for that to happen more data (soils, precipitation, probability of drought, etc.) that would strengthen the hypothesis testing and that will act as controls need to be brought to the analysis and tested with model selection. For example, can you get phytolith data from non-Si accumulating species across the same MAT gradient? I think you can, and it would be the appropriate test of your hypothesis. In fact, because you have an explicit hypothesis and expectation that Si accumulators increase Si in relation to MAT and not species that are not Si accumulators, I think this analysis is only robust and convincing if you also test the relationship for non-Si accumulating species. What if the non-accumulators show the same relationship – that weakens your hypothesis.

Fig. 3b and Fig. 3d – How are individual samples represented in these graphs? Are these sites or individuals? The sample size suggests there are 475 individuals sampled, but are there 475 points on graph b? And if not, how many sites and how are you handling the nested analysis? This should probably be a correlation (not a glm) based on the analysis of site level averages (so n should equal the number of sites not individuals). Based on this analysis the confidence band on the slope is inappropriate. You can leave the line but reporting the Pearson's r (based on site averages) tells the reader everything they need to know. Finally, I don't understand the significance of Fig. S3 – why are these two regions analyzed separately? Because they didn't fit on the line? But if that is true then it suggests something else other than temperature (like soils or water) also controls phytolith concentration and should go in your model! The authors are covering up important aspects of the data here.

Lines 117 – 121: much more information is required on the transport proteins, evolutionary tree construction and the BLAST search.

Lines 121 – 124: what does this mean about more taxa? How many species and how were they constructed. This line does not instill much confidence in the results.

Line 139 & Fig. 4: lots to say about this figure. First, the figure is difficult to read because the highlighted red periods are barely visible. Second, is there a way to expand temperature across the entire width of the figure so you can put all four phylogenies underneath the temperature fluctuations? The version of the figure is very difficult to read because Lsi3 and Lsi6 are off to the side, and it took me time to figure out what I was looking at (also because the hot periods in red are not visible). Also, how were the warm periods identified? There are periods that are cool (if not cold) relative to other periods which you have identified as warm – how, on what basis? Your definition seems arbitrary at best. However, if there is a systematic paleo definition of warm periods based on established literature that would be the best approach. Alternatively, maybe it is the relative temperature (i.e. compared to the recent past) that matters, but if so, you need to provide a clear methodological justification.

Lines 160 – 165: again, why these five subfamilies? How were they chosen and why not others?

Line 284: why surface-sterilized with 10% hydrogen peroxide for 10 minutes?

Line 290: how many seedlings were selected out of how many seedlings in total.

Line 302: this is confusing, do you mean five replicate plants (like true replicates)? The way it is worded it is unclear how many true replicates exist... also because we don't know the sample size of the experiment, which is crazy. If it is five subsamples per plant for example the entire experiment is bunk.

Line 311: how was the mean and standard error calculated – from all individuals sampled within these taxon groups? I realize these data are from another paper but please include sample sizes. Even better would be to provide the raw data so a reader could rebuild the dataset if they wanted. Are these the data from Table S3, because those data do not have sample sizes or standard errors reported.

Lines 313 – 318: We need more information on why these plant families with either $> 2 \text{ mg g}^{-1}$ or $< 1 \text{ mg g}^{-1}$. Are these values chosen arbitrarily by the authors or are these standard high and low values. Is there a natural cutoff point or statistical reason for choosing these values?

Lines 327 – 328: I will repeat this comment which I also made earlier in the manuscript. This analysis requires a control and would be made much stronger by comparing the variation of Si in response to MAT from these two species from two species from the non-Si accumulating group of plants.

Lines 338 – 340: I don't know what origin 2021 is but this is not reproducible research. This should be a correlation done in R with scripts provided. The following sentence suggests that R is used for other analysis so why not do it in R and provide the data and scripts?

Lines 346 - 348: this is a completely unacceptable amount of information provided for the phylogenetic tree building analysis and represents unreproducible research.

Lines 350 – 353: but these thresholds of above and below 18°C do not align with the shaded red regions in Fig. 4. But to be honest now I can't tell if there are supposed to be shaded red regions in the phylogenies of Fig. 4 or if they are only supposed to occur on the tips of the phylogeny (i.e., the species names).

Reviewer #3

(Remarks to the Author)

Summary

This manuscript explores the idea that high air temperatures drove the evolution of silicon use by plants – that is, that plants use Si in hotter climates, in part to manage stresses associated with heat. The authors use multiple data sets – from field collections, experiments and global datasets to test the hypothesis. It is a novel perspective, and a provocative idea. The discussion is largely well put together and likely to promote new thinking and research, and also places this work in context and explains why it matters (not least in contributing to our understanding of how and why plants use silicon, and linking global C and Si cycles, and the role of plants in the regulation of both over geological timescales). The hypothesis is somewhat supported, but the final sentence of the paper is perhaps more accurate than the title – evolution of Si in plants correlates with climate, but whether that is temperature or water/VPD driven remains unclear. There are data gaps in the global data sets (most data from mid-latitudes). The authors note (L32), some patterns found could be linked to high transpiration rather than high temperatures, and the manuscript would be substantially strengthened if this was better explored. 'Transpiration' is mentioned just once in the manuscript. Considering vapour pressure deficit (VPD) seems the most straight forward way to explore if managing water loss or temperature stress is the key driver here, but is currently missing from the manuscript. It may not be possible to retrospectively measure this for the field data collections and experiments, but adding it to global distribution analysis should be possible. There is a need to better justify or clarify some of the methods also.

Main issues

It is not always clear what aspects of plant Si has been measured. Leaf Si or plant Si? Si concentration or phytolith concentration. At times the data sets used seems too high for whole plant Si, but if this is leaf Si, then it could be presenting only part of the story. This warrants discussion.

Stromberg et al's 2016 paper is referenced, but there is no mention of the idea presented in this work that the evolution of insect herbivores could be a driver of Si accumulation in plants. I appreciate this makes the argument in this paper less elegant, but I think it is important to note that others have suggested other possible drivers given the list presented here. I was surprised not to see the inclusion of a discussion of C3 vs C4 plants, given the focus on temperature. Brightly et al. was mentioned (L325-7), but not the aspects of photosynthetic pathways. This could deepen the discussion – why might Si not be linked to this aspect of plant biology, but still driven by temperature?

The paper focuses on the idea that temperature (mean annual temperature) could be a driver of plant silicification – regionally, but also over evolutionary timescales. This can be related to water stress, no attempt has been made to exclude this possibility. It is mentioned, where it is noted that patterns linked to high temperature L32 “could also be attributed to increased transpiration accelerating passive Si absorption” then not considered for the rest of the analysis. Repeating analysis with VPD measures rather than MAT could be illuminating. If areas with high temperatures, but little drought stress

(ie. more tropical areas) did not show high Si accumulation, then the argument would be stronger for transpiration as a driver, rather than MAT.

- Stronger signals were shown in wheat than rice, and wheat is more likely to have evolved in conditions with water stress than the water-hungry rice.

- Drought can often be correlated with high temperatures, so correlations could be masking true drivers

- Figure 2 shows that the areas experiencing water limitations are least well covered either by the families selected or the data available.

Also the recent paper by Cooke and Carey (2023), which provides meta-analysis support that Si increases water movement in plants suffering drought (and salinity) stress, which would support the arguments made here, warrants a mention, and further suggests a discussion of transpiration is needed.

The authors have used multiple lines of evidence which is a strength, but this may mean their word count has been used to explain all of these, possibly the reason deeper consideration of some key issues around the main topic are missing, e.g. the potential role of transpiration. That said, I did really enjoy reading the discussion.

Specific comments

Figure 2 – y axis needs a scale, or perhaps two.

Figure 3 – why phytolith concentration instead of silicon concentration? What proportion of total plant silicon is phytoliths? When considering functions, this is important.

Figure 3d is not convincing, with an r^2 of 0.03. A high number of samples has led to the significance value rather than it representing a strong pattern. I worry that this means it reports a relationship between increased transpiration rather than Si use in hot climates. The reason rice does not show such a strong relationship is because it is semi-aquatic and transpiration rates are likely different compared to wheat.

Figure 4 – This figure is really unclear. It does not link to the text well. It would be better to separate 4a from 4b&c, as the figure title is confusing. Red is used to represent both warm periods and high-Si species, which is problematic.

Figure 4a - should be noted that all of these species are grasses. What are the implications of this? What is the value of including multiple species from within a genus, rather than looking at more genera?

Figure 4b – It is hard to link this to the next. List the families and label the climate periods perhaps?

Figure 4c – This figure is much more complicated than it needs to be. The curves on these which extend beyond the data, are not needed, nor the legend, which repeats the same information as the x axis. The box-plots with points (jitter) should be sufficient.

L78-79: This level of description (ie. defining high and low Si accumulation) is well done.

L85-87: Could consider tropics vs deserts etc here? The distribution of species seems quite dominated by some latitudes.

L95: Significant-> significant but very weak, also why phytolith concentrations? I think from L33 onwards, this is actually Si concentration, but if no, why was something different measured, and what are the implications?

L123: include -> included.

L132-135: this sentence has multiple grammar errors

160-185: It is not clear how this paragraph relates to Figure 4, if at all. This Figure, and this paragraph and the one before could be much clearer and better joined. Could put some of the terms e.g. "Middle Eocene Thermal Maximum" on Figure 4b if they are part of the same data etc.

L187: it seems odd to refer to the evolution of the climate when the paper is arguing that climate is an evolutionary driver of Si accumulation. Consider re-phrasing.

L198: Only vertebrate herbivores has been mentioned in the paper, despite Stromberg's suggestion, so this should also clarify that statement only refers to vertebrate herbivores.

L201: Yes I think the role of temperature can be stated so confidently, without exploration or discussion of water loss or water stress.

L267-8: This sentence needs more details to be meaningful. Perhaps an example, otherwise it is too vague. The previous sentence refers to cold weather preventing evolution of high Si use, but this sentence talks about the 'loss' of high Si capacity. How are the two linked? Do species change Si accumulating capacity or do families that do or don't accumulate Si flourish/expand at different times?

L275-6: "and our findings on the link between plant silicification and changes in the Earth's climate" – yes, I think there is strong evidence for this, but which aspect of climate hasn't been really clearly pinned down.

Version 1:

Reviewer comments:

Reviewer #1

(Remarks to the Author)

I read your replies to my previous evaluation of your manuscript and appreciated that you have accepted the comments. After careful reading of all your responses to other reviewers and the new version of the manuscript I add just a few additional notes.

Generally, the text was considerably improved, and many new data were included. However, there are some points which I recommend clarifying or modifying.

The data about the effect of silicon in case of other stresses cannot be completely neglected. So, I cannot agree with the sentence:

Page 34 ...In addition, we have specifically discussed the inadequacy of evidence regarding the impact of other hypotheses on silicon accumulation, including humidity (or drought), herbivores, habitat types, photosynthetic pathways, and CO₂ levels (lines 347-403).

You can discuss and support your opinion by your experiments, but the large amount of data showing the positive effect of Si on plants exposed to some other biotic or abiotic stresses cannot be neglected.

The other problem may be the Si quantification by extraction of phytoliths and considering it as silicon concentration of leaves – or even plants. Total plant silicon is certainly not only phytoliths. Please note the publications showing Si – cuticular layer, silica deposition in sclerenchyma cells (of leaves and stems) in many species. I understand the difficulty with the method of qualifications, but it should be noted and underlined that this may affect the results (if some comparative measurements are not possible).

And when speaking about Si in plants, do not forget about deposits of silica in roots. The data about root Si content or concentration are scarce.

Fig S 2 fresh weight – of what – shoot + root?

Reviewer #2

(Remarks to the Author)

Thank you for the opportunity to conduct a review this revised manuscript. While the results and conclusions remain relatively unchanged, the readers confidence in the results is vastly improved by the changes made to the manuscripts by its authors. I appreciate the work that the authors did to address my previous comments and make improvements to the manuscript. In particular, I appreciate the analysis of non-silica accumulating plants across the temperature gradient (Fig. 3b), the inclusion of more predictors in that analysis, and the additional experiment teasing apart the effects of humidity from temperature on silica accumulation. These improvements to the manuscript provide great insight into the mechanism of leaf silification, how it can protect a plant against high temperatures, and clarity about its evolutionary origin. So congratulations on a paper that shows the best evidence to date that global temperatures can explain the evolution of silification across plant families.

A few additional questions and comments:

Pg 6, L 140: you report that many elements were measured in the plant leaves but apparently not carbon or nitrogen. Is that correct? As these are percentages, it seems pertinent to a mechanistic understanding of silica accumulation to know how carbon is changing as well. For example, it would be interesting to know if the background carbon is staying about the same and silica is increasing or if carbon is going down while silica goes up. Obviously, this is not critical to the dataset, but if you have the data on C and N I think it would be very interesting to know from a physiological standpoint.

Pg 7, L 155: I think you mean “1,000”, right?

Pg 7, L 160-161: I don't understand the point of this sentence. Are you comparing high-Si and low-Si plants in the tropics? If so, 14.2% and 13% just don't seem that different and well within the expected range of error. Was this a typo and it was supposed to be 1.3%?

Pg 8, Fig. 2: can you tell the reader what the blue and red dots are and their associated lines. Are these means and standard errors or something else? Also, can you add standard deviations to the table in C?

Pg 9, L 191-198: you refer to mean annual temperature (MAT) and average pressure (PRS) but not means of any of the variables (like PRE, GST, EVP, etc.). Why not? Were they not means or was this just an oversight in the rewriting of this section? Please clarify how the aggregation of these variables over time differs (or does not).

Pg 12, L 260-262: this sentence is deceptive. Are 59% and 54% somehow statically different? You did a nice job above describing the evolution of the reported proteins among families, but I am wondering if simply reporting the percentages is enough. I am wondering if there is a way to create a null model based on the tree and times since divergence and then randomize the families across the origin points many times to get the expected null distribution. Then you can ask if the observed values fall outside the 95% CI for the mean of that distribution. Without some sort of null model approach, I am not sure these percentages provide much insight.

Pg 15, L 338: I suggest you change the sentence to read “...lignin, and therefore may damage...”

Pg 15, 341-343: are “deep undercooling” and “deep supercooling” different? If so, explain. If not use the same phrase.

Pg 17, L 392-398: I suggest you strengthen your wording here and say that while there is evidence that other factors adjust plant Si through plasticity the current analysis provides the strongest evidence to date that global temperatures provide an evolutionary explanation for the rise of plant silification. Don't shy away from your results and strong evidence – showcase them!

Pg 17, 401-410: same comment here. You spend the entire paper building a really strong case and then you cite a bunch of correlative studies that tended to focus on one or few factors. Citing paper with different results is important but don't do all that work and then say that you may be wrong and silification could have evolved for X, Y, and Z reasons.

Reviewer #3

(Remarks to the Author)

This manuscript explores the idea that high air temperatures drove the evolution of silicon use by plants – that is, that plants use Si in hotter climates, in part to manage stresses associated with heat. The authors use multiple data sets – from field collections, experiments and global datasets to test the hypothesis. It is a novel perspective, and a provocative idea. The discussion is well put together and likely to promote new thinking and research, and also places this work in context and explains why it matters (not least in contributing to our understanding of how and why plants use silicon, and linking global C and Si cycles, and the role of plants in the regulation of both over geological timescales). Substantial evidence is provided to support the hypothesis, but other possibilities and limitations are well explained. The careful consideration of the consistent reviewers comments has strengthened the manuscript considerably, and I found the arguments clear and convincing now.

The data used for Figure 2 comes from Hodson's data which was placed on a relative scale. If the raw data is plotted (Figure attached shows relative Si and raw Si data comparison) the high and low clades change. I think it would be worth repeating the analysis using the ordering from the raw data. The pattern may better support the hypothesis?

I have a few other small recommendations:

L1: for temperature-dependent -> for the temperature-dependent

L155: 1.000 -> 1,000

L161: only 13% -> 13% (I think if it was 40% compared to 13% you could use 'only' but a difference of 1.2 percentage points is not huge).

L264: GAT needs an explanation.

L318: Delete: typical

L328: cell wall -> cell walls

L350: When analyzing the mechanism by which temperature affects plant Si uptake, it is usually disturbed by humidity, ->

When analyzing the mechanism by which temperature affects plant Si uptake, it is usually confounded by effects of humidity,

L351: "as temperature not only affects energetically expensive Si transporters like Lsi2" I don't understand the logic of including this sentence, and how it fits with the rest of the argument.

L350-353: This is a long and confusing sentence, and how it fits with the next sentence isn't completely clear either. It would be worth re-writing as it is a key argument, possibly several arguments.

L373-375: References needed to support this claim.

L375-378: These are not plants

L401: Not ideal to start a paragraph with 'for example'.

Figure 3 b, c, d: the writing on the axis titles is very small. It might be better to have a longer figure that is legible.

Figure 3 c, d, e, f: Please put parameters in the same order.

Figure S6. Arrows have been added where they fall in warm periods, I don't think they are show where that is not the case. I think it would be fairer to show all arrows, at let the reader decided how convincing the results are. I think it will be a figure that still contributes support, but is more transparent in method.

(Attachment at end of manuscript.)

Response to Reviewers

Manuscript No.: NCOMMS-24-18614

Title: Convergent evidence for temperature-dependent emergence of silicification in terrestrial plants

Dear Reviewers,

Thank you for your efforts in reviewing our manuscript and helping us improve the quality of the paper. We greatly appreciate your professionalism and dedication.

In response to your suggestions, we have made significant enhancements to the manuscript. We conducted two additional experiments to investigate the effect of Si on the response of upland rice to high and low temperature stress, as well as the effect of temperature on Si accumulation in rice under the same humidity. Additionally, we have conducted sampling on two non-silicon accumulating plants in China (weeping willow [*Salix babylonica* L., Salicaceae] and winter jasmine [*Jasminum nudiflorum* Lindl., Oleaceae]) and analyzed the relative importance of nine climate variables, including mean average temperature (MAT), vapor pressure deficit (VPD), in explaining changes in leaf phytolith concentration of high-Si plants. We have also rewritten the results of the manuscript based on your suggestions, providing detailed experimental steps and all original data and codes to improve overall text quality and reproducibility. For the discussion section, we have added discussions on other hypotheses for variation in Si concentration, such as humidity (or drought), herbivores, and CO₂ levels, to highlight the important role of temperature in plant silicification. In addition, we have updated the references to reflect the latest research developments in the field.

We believe that these revisions provide more evidence and a better reading experience for the audience. The knowledge presented in this manuscript regarding the relationship between silicon-accumulating plants and temperature is valuable to a wider readership, contributing to a better understanding of plant silicon biology and its implications for global climate change.

Comments from Reviewers:

Reviewer #1:

1.The manuscript submitted to Nature Communications represents valuable contribution discussing and critically reviewing the knowledge of Si accumulating plants in relation with temperature. The authors analyse the distribution of these species from the viewpoint of systematic botany and historical phylogenetical development of plant species. The results are surprising and future discussion about this topic can be expected. The authors add results of one experiment in addition to the review of present knowledge.

Thank you for your positive evaluation of our manuscript. We are pleased that you recognize the value of our contribution in discussing and critically reviewing the knowledge of silicon-accumulating plants in relation to temperature. Your appreciation of our approach, which combines a systematic botanical perspective and historical phylogenetic analysis of plant species, is greatly encouraging. By reviewing the present knowledge, providing contemporary plant distribution evidence, and incorporating control experiments, we aim to offer a comprehensive and multifaceted assessment of the relationship between silicon-accumulating plants and temperature. Your feedback reinforces our belief that this work will inspire further research and we are committed to contributing to the understanding of plant silicon biology and its implications for global climate change.

2.The text is generally well written and easy to follow. I have just a few comments and suggestions to add some data for extending the problem more broadly in relation with drought often accompanying climate warm periods and may be some additional data about Si accumulating species can be mentioned.

Based on your recommendation, we have supplemented the manuscript with additional experiments and discussion to better understand the relationship between humidity/drought and plant silicification. We have conducted an additional experiment to differentiate the effects of temperature and humidity on Si accumulation. Then, we analyzed the relationship between the leaf phytolith concentration in both wheat and rice (high Si species, as in the original paper) and nine climate variables. These climate variables not only include mean annual temperature (MAT) (as in the original paper), but also now include vapor pressure difference (VPD), average ground surface temperature (GST), evaporation (EVP), precipitation (PRE), average pressure (PRS), average

relative humidity (RHU), average wind speed (WIN), and surface downwelling shortwave flux in air (RSDS). In addition, we specifically discussed the role of Si in alleviating drought stress in various Si accumulating species, as well as the possible relationship between humidity/drought and plant silicification. We believe these additional experimentations and analyses makes our conclusions more robust, as we explain in our response to reviewer 2 (see below), where we provide details on the results and conclusions from these additional experiments.

3.The authors do not mention several publications dealing with the role of Si cycle regulating the Earth's carbon cycle e.g.: Hodson, M.J., Guppy, C.N. Some thoughts on silicon and carbon trade-offs in plants. *Plant Soil* 477, 233–239 (2022). <https://doi.org/10.1007/s11104-022-05394-5>

Thank you for bringing the valuable work of Hodson and Guppy to our attention. We agree that their paper provides fascinating insights into the silicon and carbon trade-offs in plants, and we have incorporated their findings into our revised manuscript to expand our introduction (lines 19-22). In addition, we have included a discussion of the key hypotheses presented in their opinion paper (lines 350-353 and 393-395).

4.Some additional data about experiments confirming the role of Si for sorghum and sugarcane exposed to drought can be mentioned or discussed in the chapter Discussion.

Thank you for your insightful suggestion. As you point out, our original manuscript lacked discussion on the role of silicon in non-rice crops, which could be significant given that rice often grows in paddy fields whereas many other Si-accumulating crops grow in drier conditions. Our revised manuscript notes that Si has the ability to help sorghum and sugarcane, as well as rice, tolerate drought stress (lines 358-361), and based on your suggestion, we have emphasized drought as one important factor that may drive plant Si accumulation in rice and non-rice plants in our discussion (lines 358-367).

5.Rice (*Oryza sativa* L. cv. Nipponbare) was selected for the experiment testing the effect of Si on high and low temperature stress. This cultivar, often used for experiments is a typical lowland cultivar. Comparison (of Si accumulation or even growth reaction) with some upland cultivar adapted to different water and temperature conditions may bring some new aspect to the

present contribution. The data about the difference of Si accumulation in lowland and upland rice cultivars are known from the literature.

This is a valuable suggestion. Comparing the silicon accumulation and growth response of upland rice varieties adapted to different water and temperature conditions with lowland cultivars like Nipponbare would indeed bring new insights into our study, so we have carried out another set of experiments to test this. We have now conducted studies (lines 68-77, and 444-451) on rice varieties involving lowland and upland rice, including indica, cv. Jinzao 47 (lowland rice), japonica, cv. Linhan 1 (upland rice), japonica, cv. Danhan 53 (upland rice), and japonica, cv. Zhenghan 10 (upland rice). We cultivated 10-day old rice in nutrient solution with/without 1mM silicic acid for 15 days before stress. Then, the rice was subjected to high temperature (40 °C) and low temperature (0 °C) stress. Three days later, we took photos and measured shoot length, root length, fresh weight, and chlorophyll concentration. Results also showed that Si has the ability to help these other rice varieties tolerate high temperatures (40°C), significantly increasing shoot length, root length, fresh weight, and chlorophyll concentration. However, Si was harmful to them under low temperature (0°C), manifested by more severe wilting and a decrease in various indicators (new Fig. S2).

In addition, we have amended the discussion to highlight studies of the relationship between drought and plant Si concentration and papers that compare Si concentration in upland rice and lowland rice (lines 362-364). In summary, our assessment of the literature, including studies from both field and laboratory settings indicates that compared to temperature, drought is not the main cause of silicon accumulation in plants and we have incorporated this discussion into our manuscript (lines 358-367).

6. Specific comments

Fig. 3 MAT - better write: mean annual temperature – it will be more clear for the reader.

We have revised the horizontal axis label in Fig. 3 from “MAT” to “Mean annual temperature” to improve clarity. In addition, this figure has been thoroughly updated with the addition of two new low-Si species and analysis of more climate variables.

Reviewer #2:

1. **** Major Comments ****

This manuscript addresses the evolutionary origins of plant silicification, an important, timely, and moderately controversial topic in plant evolution and ecology. There is a lot to like about this manuscript and I appreciate the multiple sources of data that were used to address the question of why plants have evolved to concentrate silica in their tissues. There are literally hundreds of studies which have quantified silica concentration in plants and dozens which have proposed ecological and evolutionary drivers of plant silicification. Special topic papers or reviews in high profile journals over the last ten to fifteen years highlight the lack of consensus on why some plants concentrate Si in their tissues and others do not. Three issues have traditionally plagued these studies: (i) the various proposed drivers of plant silicification are often correlated and difficult to tease apart, (ii) attempts to answer this question have been largely observational and have lacked experimental rigor, and (iii) the hypotheses underlying these explanations have lacked a clear mechanistic evolutionary explanation (i.e., an ultimate rather than proximate cause).

This paper attempts to overcome points (ii) and (iii), with less of an attempt to overcome point (i).

Past studies that have failed to address point (i) have focused on one mechanism and have sought to provide evidence for that single mechanism, rather than trying to eliminate alternative competing mechanisms that may be correlated but not ultimately responsible for the observed increase in Si. Right now, there is a long list of potential reasons Si accumulation may have evolved in plants. This remains an important setback for this paper. In fact, parts of their analysis, which I address below, seem to cover up the fact that in observational studies other important environmental drivers, such as soil or potentially rainfall, contribute to observed phytolith concentrations in crop systems. To me a goal of this study, and worthy of publication in a distinguished journal like Nature Communications, should be to eliminate alternative mechanisms in favor of temperature being the principle and ultimate evolutionary factor responsible for plant silica concentrations in deep time. To do this will take some work because it will mean bringing other, potentially correlated, variables into the analysis and asking which is the best explanatory

variable. If you cannot eliminate those other potential variables, then all you are doing is throwing one more variable onto an already large pile of possible explanations. I have three major suggestions for this manuscript:

Thank you for your thorough and insightful review of our manuscript. We are grateful for your recognition of the importance and timeliness of our research on the evolutionary origins of plant silicification. Your in-depth understanding of the current state of the field and the challenges faced by previous studies is greatly appreciated.

Based on your valuable feedback, we have supplemented the control experiments, conducted new national-level sampling, collected additional climate data, and specifically addressed several other hypotheses in the Discussion section. We have made a concerted effort to refute alternative factors and provide stronger evidence for the temperature-driven plant silicon accumulation hypothesis.

2. My first major suggestion for this paper is to bring other variables that can account for variation in plant silicification into the observational regression analysis presented in Figures 3 and S3. Typically, I try to review papers based on the work done, rather than on work that was not done. However, in this case it is essential to the nature of the argument that other factors be discussed and possibly eliminated as explanations for plant silicification. Consequently, other environmental factors need to be included in the observational analysis of phytolith concentration across China and a model selection procedure should be used to identify the best explanation for the response. Second, across the MAT gradient in China I urge the authors to find a parallel dataset for no-Si accumulating crop species. Implicit in their hypothesis that evolutionary periods of high temperature are responsible for Si-accumulation in plants is that plants which evolved during cool periods should not respond to gradients of temperature by altering Si concentrations in their tissue. This can and should be explicitly tested by including an analysis of non-Si accumulating plants across the same gradient of MAT.

Thank you for your valuable suggestions. We accepted your advice and have made significant efforts to address the concerns you raised.

Firstly, we have taken up your suggestion to include two parallel datasets for non-Si accumulating species across the same mean annual temperature (MAT) in China. This is a crucial point, as our hypothesis that evolutionary periods of high temperature are responsible for Si-accumulation in plants implies that plants that evolved during cool periods should not respond to temperature gradients by altering Si concentrations in their tissues. Over the past few months, we have collected samples from two types of non-Si accumulating plants (weeping willows and winter jasmine) in China (new Fig. 3a and b). By comparing the response of Si-accumulating and non-Si-accumulating plants to the MAT gradient, we can explicitly test our hypothesis and provide stronger evidence for the role of temperature in driving plant silicification (lines 130-144).

Secondly, we have included more climate variables in the observational analysis of leaf phytolith concentration of high-Si plants across China. In addition to mean annual temperature (MAT), we collected additional variables on climate such as vapor pressure difference (VPD), average ground surface temperature (GST), evaporation (EVP), precipitation (PRE), average pressure (PRS), average relative humidity (RHU), average wind speed (WIN), and surface downwelling shortwave flux in air (RSDS) to investigate their potential influence on plant silicification (new Fig. 3c-f). By incorporating these factors into our regression analysis, we aimed to identify the best explanatory variable for the observed patterns of plant silicification. We have used model selection procedures to determine the relative importance of each variable and to assess whether temperature remains the most significant driver of plant silicification when other factors are considered.

We believe that these additional datasets and analyses will significantly strengthen our study and provide a more comprehensive understanding of the evolutionary drivers of plant silicification. By addressing the concerns you raised and incorporating a wider range of plant types and variables into our analysis, we aim to provide a more robust and convincing argument for the role of temperature in shaping plant silicification over evolutionary timescales.

3. My second major comment is to provide better justification for some of the decision points in the manuscript which can be constructed as arbitrary. These include (i) the thresholds of high ($>2 \text{ mg g}^{-1}$) and low ($<1 \text{ mg g}^{-1}$) silica concentrations.

Thank you for raising these important points regarding the decision-making processes in our manuscript. We acknowledge that providing clear justifications for our choices is crucial for the robustness and transparency of our study. We have carefully reconsidered and redefined several decision-making approaches based on your comments.

Firstly, regarding the thresholds for high ($>10 \text{ mg g}^{-1}$) and low ($<1 \text{ mg g}^{-1}$) Si concentrations, we recognize that there is no unified standard for classifying plants based on Si concentration. We have reviewed the literature and found that various researchers have proposed different classification schemes. For example, Jones & Handreck proposed a rough division of plants into three groups based on their dry weight-based silicon content: wetland gramineae have the highest values, of the order of 10-15% on a dry weight basis; dryland grasses such as rye and oats are intermediate about 1-3%; and most dicots have less than 1%¹. Ma & Takahashi used both Si content and Si/Ca ratio as criteria for Si-accumulating plants and non-accumulating plants². Plants with a Si content and Si/Ca ratio higher than 1.0% and 1.0, respectively, are defined as Si accumulators. By contrast, plants with a Si content lower than 0.5% and Si/Ca ratio lower than 0.5, are defined as Si excluders (non-accumulating plants). Plants with a Si content and Si/Ca ratio in between Si accumulators and excluders are intermediate type plants. In addition, Coskun et al. proposed a different classification system³. Active accumulators have a leaf Si content ranging from 1.5% to 10%. The Si content range of passive accumulators is from 0.5 to 1.5%. The rejective classification applies to plants with a Si content of $< 0.2\%$.

In this manuscript, we have redefined our thresholds based on these existing classification methods. We now refer to families with silicon concentration $>10 \text{ mg g}^{-1}$ (1%) as high-Si families and those with silicon concentration $<1 \text{ mg g}^{-1}$ (0.1%) as low-Si families. This division is intended to express the relatively high or low Si concentration of plants from different families for the purpose of evolutionary analysis. We have clarified the selection criteria for this paper in the main text and supplementary materials (lines 194-197) (Fig. S7).

4.(ii) taxonomic analysis (i.e., which plant families or subfamilies were included in the analysis).

Specific information on high- and low-Si families used for evolutionary analysis has been

added to the figure (new Fig. 4a) and the supplementary materials (Table S9). We have published detailed data on the leaf silicon concentration of 1826 plant species and 213 families in the supplementary materials (new Table S7), to facilitate reanalysis by readers as needed. Within the same family with a large number of species, there are also significant differences in Si concentration among plants from different sub-clades (e.g., subfamilies within a family). In order to provide more evidence for the hypothesis that temperature drives the evolution of plant Si, we selected five families with the highest number of species among angiosperms to separately analyze the evolutionary time of their different sub-clades (Table S10). There are five plant families containing more than 10,000 species each and, thus, they are recognized as the largest families within the higher plants in terms of species diversity, namely Asteraceae, Orchidaceae, Fabaceae, Rubiaceae, and Poaceae^{4,5}. We have also added selection criteria and relevant literature in the main text (lines 231-237).

5.(iii) the temperature cutoffs for warm and cool periods (18°C and which lineages arose in warm or cool periods)

Based on lithologic evidence and oxygen isotope methods, Scotese et al. divided the periodic fluctuations of Earth's historical temperature into warming and cooling episodes, mainly reflecting trends of temperature change, and it is difficult to have a strict critical point (nearly 500 citations)^{6,7}. Christopher R. Scotese is a renowned paleogeographer, whose maps and animations depicting the evolution of the Earth over the past billion years are widely used in textbooks, research papers, and global museums (<http://www.scotese.com/>, <http://54.189.216.145/GlobalGeology/>). Our classification of warming and cooling episodes (new Fig. S8) has been redrawn based on his paleotemperature timescales (Fig. 15 and 19 in Scotese's paper) and the published raw data⁷. To ensure the rigor of our manuscript and provide readers with a better reading experience, we have added descriptions in the main text (lines 197-201, 552-557), and included temperature data of the ancient Earth in the supplementary materials (new Table S8). In addition, the statistical test results of the emerging time of high- and low-Si families in Fig. 4b can provide more direct evidence, solving the confusion caused by the lack of a critical point in the division of warming and cooling episodes.

6. My third major comment is to provide better descriptions of the methods, statistical analyses, and results. I urge the authors to add much more information and detail into future versions of this manuscript. Telling readers "... rice plants were placed under three types of stress (heat, cold, and freeze thawing) for three days" is not detailed enough for an outlet like this. Papers I have read and written for these condensed format journals are packed with information, and for good reason – space is limited. But as it currently stands there is nowhere near enough information provided for readers to understand the analysis or judge the validity of what was done. The lack of information about basic methods and results, which I comment on in detail in the *Specific Comments* section, is not at a level that is acceptable for a peer-reviewed outlet. I understand well the "iceberg" analogy for papers formatted for science and nature, but that does not mean it is acceptable to leave essential information out of the "tip of the iceberg" which is published in the main journal. To continue this analogy, the submerged part of the iceberg (i.e., the supplemental materials) in this manuscript is a disappointing fourteen pages including several long tables filled with information about various plant families. The content presented in the methods and supplemental materials are insufficient and do not currently represent reproducible research.

There are several specific issues with the presentation of results, data, and statistics. Most (but not all) of these issues can likely be dealt with by providing more information. The manuscript lacks basic reporting of statistical tests and associated test statistics and the assumptions (or violations) of statistical tests that were used. My suggestion is that any time you report results tell your reader what analysis was used to arrive at that result along with any associated test statistics or diagnostic information. Then refer the reader to the methods or supplementary materials where you can provide all the details of the test and exactly how it was carried out – to the point that the reader could replicate exactly what was done if they so choose. The paper needs to be re-written to include all the methodological details of the paper in expanded form, either in the methods or in supplemental materials. Moreover, I urge the authors to consider including data and scripts so that all analyses are reproducible. Publishing fully reproducible research with code and data to rerun all the analyses is the gold standard for scientific publishing for a journal such as Nature Communications. This manuscript has a long way to go.

Thank you for your thorough and constructive comments on our manuscript. We appreciate

your guidance on the need to provide more detailed descriptions of the methods, statistical analyses and results, in order to meet the standards of a high-quality journal like Nature Communications. We have made significant efforts to address your concerns and improve the manuscript. Based on your suggestions, we have expanded the information provided in the methods section and supplementary materials to ensure that our research is reproducible and transparent.

Specifically, we have added more details about the experimental procedures, such as the stress treatments applied to the rice plants. We now provide comprehensive information on the setting of biological replicates, selection of silicic acid concentration, and the specific conditions of the heat, cold, and freeze-thawing treatments both in the main text (lines 59-60, 413-454) and the figure (Fig. 1 and S2). We also provided a detailed explanation of why H₂O₂ was used to soak seeds for disinfection (lines 415-417), and how to convert SPAD values into chlorophyll concentrations (lines 439-441). As another example, we also provide protein sequences of species and pairwise divergence time used to construct evolutionary trees (new Table S5 and S6), as well as detailed steps and software (lines 178-193, 536-551). These will allow readers to better understand and evaluate the validity of our experiments.

Regarding the presentation of results, data, and statistics, we have made several improvements. We have replaced the software used for regression analysis and uniformly used R for modeling. We now consistently report the statistical tests used, along with their associated test statistics and diagnostic information, whenever we present a regression result (for example, new Table S4). We have also expanded the methods section and supplementary materials (including 9 figures, 10 tables) to include all the methodological details necessary for readers to replicate our analyses. In addition, to further enhance the reproducibility of our research, we have included all the analysis code (Code S1-3) used in R. This will enable readers to verify our results and build upon our work, adhering to the gold standard for scientific publishing in top-tier journals.

7.** Specific Comments **

Abstract: define “high temperature stress”, “warm geological periods” and “cold geological periods” in the abstract. These relative terms really need to have values associated with them to be

meaningful.

We have revised the abstract to include clear definitions of the terms.

8.Lines 55 – 56: the treatments need to be stated explicitly here.

We have provided a detailed description of three types of stresses.

9.Lines 60 – 61: is the intention to relate the freeze thawing to chlorophyll concentration? If so, you need to tell your reader explicitly what you want them to know – don't make them connect the dots on their own.

Yes, the chlorophyll concentration in the +Si group was significantly lower than that in the -Si group under freeze thawing. We have revised the manuscript to clearly state the differences in shoot length, root length, shoot fresh weight, and chlorophyll concentration between the +Si group and the -Si group under freeze thawing conditions.

10.Line 62 & Fig. 1: in panel (d) should the label be related to Si concentration? No one is going to know what SPAD means here. It should be replaced with something meaningful.

We have revised the label in Fig 1, panel (d) to "Chlorophyll concentration (mg/g fresh weight)" to clearly indicate the content of the subfigure. The SPAD-502 chlorophyll meter is a widely used tool for estimating leaf chlorophyll content in both research and agricultural settings^{8,9}. The SPAD value obtained is significantly correlated with the chlorophyll concentration ($R^2 > 0.9$)¹⁰. To facilitate readers' understanding, we have converted the SPAD values to chlorophyll concentration using an empirical formula specifically designed for the rice cultivar Nipponbare. The formula is as follows: Chlorophyll concentration (mg/g fresh weight) = $0.965 \exp(0.036 \text{ SPAD}) - 1$ ¹¹. We have also added a detailed explanation of the measurement methods and the conversion formula in the Materials Methods section, along with the appropriate references.

11.Lines 64 - 67 & Fig. 1 legend: even though the temperature treatments are located (inconspicuously) in the figure the specific treatments should be clearly indicated here.

We have revised the text to explicitly state the temperature treatments used in the study (lines 59-60). Additionally, we have updated the legend of Fig. 1 to include the specific temperature

treatments (lines 82-83). Furthermore, we have increased the font size of the temperature treatments in Fig. 1a to make them more prominent and easily readable.

12.Line 74: this is what I mean about the lack of details in this manuscript. Tell your reader the exact number of species.

Thank you for bringing this important issue to our attention. We have revised the text to state the precise number of species (lines 194-197). Additionally, we have carefully reviewed the entire manuscript and addressed similar issues where specific details were missing. We have made efforts to provide exact numbers, values, and references throughout the text to improve the clarity and transparency of our work. Furthermore, to support our findings and enable readers to access the original data, we have included the leaf silicon concentration data for all 1,826 species and 213 families in the supplementary materials (Table S7) and the statistical and plotting code used in R (Code S3).

13.Lines 78 – 83: how did you arrive at these orders? What was the procedure for selecting these orders for analysis and what justification can you provide for the focus on these taxonomic groupings.

Our selection of the high-Si and low-Si clades was based on the seminal work by Hodson et al. (2005)¹², which has been widely cited (over 800 citations) and is considered a classic study on the phylogenetic variation in plant silicon composition. In their research, Hodson et al. specifically analyzed the differences in silicon concentration across different plant clades. We fully referred to their results and selected the top 10 clades with the highest Si concentrations as the “high-Si clades” and the bottom 10 clades as the “low-Si clades” for our subsequent analysis. This approach allowed us to focus on the most extreme cases of Si accumulation and exclusion, providing a clear contrast for our investigation. To clarify our selection process, we have revised the main text (lines 101-104) as well as new Fig S4 and the associated legend.

14.Line 88 & Fig. 2b: is the average the correct metric by which to compare these distributions? What statistical test did you use for this comparison – again, the authors have not provided sufficient information to understand what was done or to judge the validity of the

analysis. The legend only tells us that there is some statistical difference at $p < 0.001$ with no information anywhere on the test, test statistics, degrees of freedom, or assumptions of the test (i.e., normality). This is unacceptable for a journal of this stature. The sample sizes in this analysis are enormous (~10,000!), meaning if they did use a parametric statistical test like a t-test then very small differences in the means would be deemed statistically significant. As a quick demonstration: two randomly generated normal distributions each with $n=10,000$ and means of 14.9 and 15.1 (assuming a standard deviation of 5, thereby producing a range similar to your temperature values) will be statistically significantly different at $p < 0.001$ approximately 32% of the time. I don't think many biologists would argue that mean temperature values of 14.9 and 15.1 are meaningful differences! And this demonstration was done with normally distributed errors. The error distribution of your data appears not to be normally distributed, making the error even worse. The information we need is about the values of the central tendency and what is the effect size of their difference. Or in this case perhaps it is more appropriate to compare distributions with a non-parametric test or analyze modes, which appear at different temperatures for High-Si and Low-Si.

We have updated the statistical method to use Wilcoxon rank sum test, which does not require the assumption of a normal distribution, to compare distribution temperature of high- and low-Si plants. We have added the methodology of the statistics and the detailed variables of the results (lines 114-119). In addition, we have followed the reviewer's suggestion and used the median and mode of the distribution temperatures of high- and low-Si plants in order to compare the differences between them in a more statistically robust way, given the error distribution. This is now included as new panel c in Fig 2.

15.Lines 93 – 97: I said in my major comments that I do not like to evaluate a manuscript based on what was not done, but in this case, it seems pertinent because (i) the many other potential mechanisms which have been proposed in the past and (ii) the bold claims this paper is making about temperature as the major evolutionary force. Most ecologists interested in this subject would love to see a really strong and definitive case for an answer (me included), but for that to happen more data (soils, precipitation, probability of drought, etc.) that would strengthen the hypothesis testing and that will act as controls need to be brought to the analysis and tested

with model selection. For example, can you get phytolith data from non-Si accumulating species across the same MAT gradient? I think you can, and it would be the appropriate test of your hypothesis. In fact, because you have an explicit hypothesis and expectation that Si accumulators increase Si in relation to MAT and not species that are not Si accumulators, I think this analysis is only robust and convincing if you also test the relationship for non-Si accumulating species. What if the non-accumulators show the same relationship – that weakens your hypothesis.

We have addressed the issue of MAT/Si relationships in non-Si accumulating species in response to your major comments (comment 2).

One of the other potential confounding factors in our analysis relates to the fact that while the sampled data suggest the importance of temperature is greater than that of humidity in Si accumulation to high-Si plants, we acknowledge that the two variables of temperature and humidity have not been directly separated. To address this, we have now conducted experiments in plant culture incubators, controlling the humidity at 70% using humidifiers and setting two temperature treatments: 26 and 28 °C (considering the optimal temperature range for rice growth is 25-30 °C^{13,14}). The methods for this new experiment are described in lines 456-478 and the results are presented in a new figure (Fig S3 a, b, c) and described in lines 89-98).

These additional experiments and analyses help to further elucidate the relationship between temperature and silicon accumulation in plants, while controlling for the potential confounding effects of humidity. We believe that these additions strengthen our manuscript and contribute to a more comprehensive understanding of the topic.

16. Fig. 3b and Fig. 3d – How are individual samples represented in these graphs? Are these sites or individuals? The sample size suggests there are 475 individuals sampled, but are there 475 points on graph b? And if not, how many sites and how are you handling the nested analysis? This should probably be a correlation (not a glm) based on the analysis of site level averages (so n should equal the number of sites not individuals). Based on this analysis the confidence band on the slope is inappropriate. You can leave the line but reporting the Pearson's r (based on site averages) tells the reader everything they need to know.

We appreciate your suggestions for improving the clarity and accuracy of our manuscript. Regarding the sample representation, we apologize for any confusion caused by our previous description. To clarify, each sample in our study comes from an independent plant individual from different locations. Although some samples come from the same village or town and have similar climate data due to their proximity, there are differences between individuals. Using the average value of the sampled plants may mask these differences between individuals and lead to inaccurate results. Therefore, we believe that using a larger sample size better represents the population and avoids bias.

17. Finally, I don't understand the significance of Fig. S3 – why are these two regions analyzed separately? Because they didn't fit on the line? But if that is true then it suggests something else other than temperature (like soils or water) also controls phytolith concentration and should go in your model! The authors are covering up important aspects of the data here.

For wheat, we have incorporated the data from northern Hebei into the overall dataset. However, the rice samples in Xinjiang are upland rice, while samples from other regions are lowland rice. There is a huge difference in water demand between the two types of rice. Also the two planting methods of paddy and dryland strongly affect soil properties and element availability¹⁵. Therefore, comparing them together is not appropriate. In order to reveal the relationship between temperature and plant silicon content more comprehensively, we did not delete these data, but conducted a separate analysis to provide more evidence for the impact of temperature on plant Si accumulation, which is presented in a new figure, Fig S5. In addition, as we have already described, we have included indicators related to water conditions such as evaporation, precipitation, average relative humidity, and vapor pressure deficit in the multiple regression model of wheat and rice (new Fig. 3). To address the ambiguity in the previous manuscript, we have provided a detailed description in the text and supplementary materials.

18. Lines 117 – 121: much more information is required on the transport proteins, evolutionary tree construction and the BLAST search.

In response to your comment, we have made the following changes to the manuscript:

(i) We have expanded the description of the Materials and Methods section to include more details on the transport proteins, evolutionary tree construction, and the BLAST search (lines 177-193, 536-551).

(ii) We have added the raw data of the BLASTP search and the pairwise divergence time used to construct the timetree to the supplementary materials (new Table S5 and S6).

However, it is important to note that due to the high conservation of the Lsi series proteins, almost all homologous protein produced by the BLASTP search using the rice Lsi proteins belong to the Poaceae family. As a result, the genetic evolutionary analysis cannot provide us with supporting information beyond the Poaceae family (lines 191-193). After careful consideration and discussion among the authors, we believe that it is more appropriate to present this information in the supplementary materials (Fig. S6) rather than in the main text. We have come to this decision because the genetic evolutionary analysis was not the core argument supporting our hypothesis in the previous manuscript.

19.Lines 121 – 124: what does this mean about more taxa? How many species and how were they constructed. This line does not instill much confidence in the results.

Upon further consideration, we agree that the analysis of transporters only provides a perspective on the evolution of temperature-related transporters and does not offer strong support for our hypothesis. To address this issue, we have decided to simplify the description of transport protein evolution analysis in the manuscript. We have removed the detailed discussion of this analysis from the main text and have not included it as part of the core evidence supporting our hypothesis. Instead, we have focused on presenting the strongest and most compelling evidence that directly supports our main argument. This includes the experimental data on the effects of Si on rice's response to high and low temperature stress and the effects of temperature on silicon accumulation, the analysis of the relationship between the geographical distribution of high-Si and low-Si plants and climate, and the phylogenetic analysis involving 17 high-Si families and 28 low-Si families.

20.Line 139 & Fig. 4: lots to say about this figure. First, the figure is difficult to read because

the highlighted red periods are barely visible. Second, is there a way to expand temperature across the entire width of the figure so you can put all four phylogenies underneath the temperature fluctuations? The version of the figure is very difficult to read because Lsi3 and Lsi6 are off to the side, and it took me time to figure out what I was looking at (also because the hot periods in red are not visible).

We apologise for the lack of clarity in the previous image. Firstly, we have split the analysis of the evolutionary tree and the evolutionary time of species into two figures and placed the figure of the evolutionary tree in the supplementary material (Fig S6). Secondly, for the evolutionary tree we have redrawn it. In the new figure, we placed all four phylogenies underneath the temperature fluctuations. Second, we removed the red shading that previously overlaid the species names and instead added information on leaf Si concentration from the database directly after the species name, and showed species with higher Si concentrations in red font. This new layout will provide a clearer and more intuitive representation of the data, making it easier for readers to understand the relationship between temperature and the evolution of the Lsi proteins. The results indicate that the Si absorption and transport proteins of species with higher Si concentrations (red font, leaf Si concentration > 20 mg g⁻¹) mainly differentiate during the warming episodes (red arrows).

21. Also, how were the warm periods identified? There are periods that are cool (if not cold) relative to other periods which you have identified as warm – how, on what basis? Your definition seems arbitrary at best. However, if there is a systematic paleo definition of warm periods based on established literature that would be the best approach. Alternatively, maybe it is the relative temperature (i.e. compared to the recent past) that matters, but if so, you need to provide a clear methodological justification.

Regarding the identification of warming episodes, we have relied on the widely recognized classification of Earth's warming and cooling episodes as reported in the literature^{6,7}. These studies, which have been cited nearly 500 times, provide a systematic paleo definition of warming episodes based on established research. The temperature map of the Earth used in our manuscript is adapted from the images and raw data provided in Scotese's paper (new Fig. S8). We have included relevant explanations in the main text (lines 197-201) and added Earth temperature information to

the supplementary materials (new Table S8) to support our classification of warming and cooling episodes.

22.Lines 160 – 165: again, why these five subfamilies? How were they chosen and why not others?

We have observed that within families containing a large number of species, there can be significant differences in Si concentration among plants from different sub-clades (e.g., subfamilies within a family). This variation prompted us to investigate the evolutionary history of these sub-clades in relation to temperature.

To select the sub-clades for our analysis, we focused on the five plant families that are recognized as the largest among higher plants in terms of species diversity, each containing more than 10,000 species, namely Asteraceae, Orchidaceae, Fabaceae, Rubiaceae, and Poaceae. By choosing these five families with the highest number of species among angiosperms, we aim to provide a more comprehensive and representative analysis of the relationship between temperature and the evolution of plant Si accumulation. Analyzing the evolutionary times of the different subfamilies within these diverse families allows us to test our hypothesis across a broad range of plant taxa and to identify potential patterns or trends that may support our argument (Table S10). We have revised the main text to include more details about selection criteria and relevant literature (lines 231-239).

23.Line 284: why surface-sterilized with 10% hydrogen peroxide for 10 minutes?

The use of hydrogen peroxide or sodium hypochlorite as surface sterilizing agents is widely practiced in seed germination studies¹⁶⁻¹⁸. In our experiment, we chose to use 10% hydrogen peroxide for 10 minutes, as this concentration and duration have been shown to effectively sterilize the seed surface without causing damage to the seeds themselves¹⁹. To provide a clearer justification for this step, we have added the relevant information to the manuscript, along with the references (lines 415-417).

24.Line 290: how many seedlings were selected out of how many seedlings in total.

In total, we germinated 300 rice seeds to ensure that we would have a sufficient number of

healthy seedlings for our experiment. From these 300 seedlings, we carefully selected 120 seedlings that were uniform in size and appearance. This selection process was carried out to minimize any potential variation in growth or development among the seedlings that could influence our experimental results. We have clarified this information in the manuscript (lines 422-427).

25.Line 302: this is confusing, do you mean five replicate plants (like true replicates)? The way it is worded it is unclear how many true replicates exist... also because we don't know the sample size of the experiment, which is crazy. If it is five subsamples per plant for example the entire experiment is bunk.

In our experiment, we used a total of 120 uniformly-sized rice seedlings, which were divided into two groups: the +Si group (treated with silicon) and the -Si group (without silicon treatment). Each group consisted of 60 seedlings, which were further divided into three subgroups of 20 seedlings each, corresponding to the three temperature stress treatments: heat, cold, and freeze-thawing.

For each temperature stress treatment, we used five replicate pots, with four seedlings planted in each pot. This design allowed us to have five true replicates for each treatment combination (silicon treatment \times temperature stress), with each replicate consisting of a single pot containing four plants.

During the measurement phase, we randomly selected one plant from each pot to collect data, resulting in a total of five replicates per treatment combination. This approach ensures that our measurements are representative of the treatment effects and not influenced by potential variations within a single pot.

To clarify this information in the manuscript, we have revised the Materials and Methods (lines 433-436, 441-443).

26.Line 311: how was the mean and standard error calculated – from all individuals sampled within these taxon groups? I realize these data are from another paper but please include sample sizes. Even better would be to provide the raw data so a reader could rebuild the dataset if they

wanted. Are these the data from Table S3, because those data do not have sample sizes or standard errors reported.

To address your concerns, we have made the following revisions to the manuscript and supplementary materials:

(i) We have included the sample sizes for each taxonomic group in Table S7 (Families). This information will help readers better understand the representativeness of the reported mean and standard error values.

(ii) We have added the raw data on Si concentrations in plant leaves to the supplementary materials (Table S7 Species). This dataset includes leaf silicon concentrations and related classification information for 1826 plant species. By providing this raw data, we enable readers to conduct their own analyses and verify our results.

We have double-checked the consistency between the data presented in the main text, Fig. S7 and Table S7. We have ensured that the sample sizes and standard errors are correctly reported in both places.

27.Lines 313 – 318: We need more information on why these plant families with either > 2 mg g⁻¹ or < 1 mg g⁻¹. Are these values chosen arbitrarily by the authors or are these standard high and low values. Is there a natural cutoff point or statistical reason for choosing these values?

We have updated the selection basis for high and low-Si clades. Our selection was based on the seminal work by Hodson et al. (2005)¹², which has been widely cited (over 800 citations) and is considered a classic study on the phylogenetic variation in plant silicon composition. In their research, Hodson et al. specifically analyzed the differences in silicon concentration across different plant clades. We fully referred to their results and selected the top 10 clades with the highest Si concentrations as the “high-Si clades” and the bottom 10 clades as the “low-Si clades” for our subsequent analysis (new Fig. S4). This approach allowed us to focus on the most extreme cases of Si accumulation and exclusion, providing a clear contrast for our investigation. We have provided a detailed introduction to the method and basis in the new version of the manuscript, and supplemented the references (lines 101-104, 480-484).

28.Lines 327 – 328: I will repeat this comment which I also made earlier in the manuscript. This analysis requires a control and would be made much stronger by comparing the variation of Si in response to MAT from these two species from two species from the non-Si accumulating group of plants.

As described in our earlier response to this issue (comment 2), we have followed your advice and expanded our analysis to include datasets for two types of non-silicon-accumulating plants (weeping willow and winter jasmine). These additional datasets serve as a control group, allowing us to assess whether the observed patterns of silicon concentration in relation to MAT are specific to silicon-accumulating plants or if they are a general trend across all plant species. The results are exciting, as the leaf Si concentration of silicon accumulating plants is significantly influenced by temperature, which does not exist in non-silicon accumulating plants (new Fig. 3b). In the updated manuscript, we have included additional text (lines 130-144) to highlight the importance of this control group and the insights gained from the comparative analysis.

29.Lines 338 – 340: I don't know what origin 2021 is but this is not reproducible research. This should be a correlation done in R with scripts provided. The following sentence suggests that R is used for other analysis so why not do it in R and provide the data and scripts?

We agree that using a widely-used and open-source software like R²⁰, instead of a proprietary software like Origin²¹, will enhance the reproducibility and transparency of our research. It will also allow other researchers to easily verify our results and build upon our work. Following your recommendation, we have made the following changes to the manuscript and supplementary materials:

(i) We have re-conducted all statistical tests, correlation analyses and multiple regression model construction using R, a widely-used and open-source software for statistical computing and graphics.

(ii) We have provided all the R scripts used for this study in the supplementary materials (Code S1-S3).

(iii) We have also included all the raw data used in the correlation analyses in the

supplementary materials (Table S3).

(iv) In the main text, we have updated the description of the statistical methods to reflect the use of R for the correlation analyses (lines 527-534).

30.Lines 346 - 348: this is a completely unacceptable amount of information provided for the phylogenetic tree building analysis and represents unreproducible research.

To address this issue and ensure that our research meets the standards of reproducibility, we have made significant revisions to the manuscript and supplementary materials:

(i) We have extensively updated the Materials and Methods section to provide a comprehensive description of the phylogenetic tree-building process. This includes detailed information on the data sources, sequence alignment methods, evolutionary models, and tree construction algorithms used in our analysis.

(ii) In order for readers to replicate our phylogenetic analysis, we have included raw data of protein sequence and pairwise divergence time for tree construction in the supplementary materials (Table S5 and S6).

We have updated the main text to clearly direct readers to the supplementary materials for the detailed methodological information and data related to the phylogenetic analysis (lines 178-193, and 536-551).

31.Lines 350 – 353: but these thresholds of above and below 18°C do not align with the shaded red regions in Fig. 4. But to be honest now I can't tell if there are supposed to be shaded red regions in the phylogenies of Fig. 4 or if they are only supposed to occur on the tips of the phylogeny (i.e., the species names).

We have made several revisions to address this issue and improve the overall clarity and readability of the figure:

(i) We have updated the main text and supplementary materials to provide a clearer explanation on the classification of Earth warming and cooling episodes based on the classic literature cited in our article (lines 197-201, Fig. S8). Table S8 now includes names of different

geological periods and more detailed descriptions of temperature.

(ii) To help readers better understand the relationship between the temperature and the evolution, we have annotated the corresponding geological period names and family names directly on the revised figure (Fig. 4a).

(iii) For the figure of the phylogenetic tree, we have removed the red shading on the species names, as it was difficult to read and potentially confusing. Instead, we have directly added the leaf Si concentration values for each species next to their names on the phylogenetic tree (Fig. S6).

Reviewer #3:

1. Summary

This manuscript explores the idea that high air temperatures drove the evolution of silicon use by plants – that is, that plants use Si in hotter climates, in part to manage stresses associated with heat. The authors use multiple data sets – from field collections, experiments and global datasets to test the hypothesis. It is a novel perspective, and a provocative idea. The discussion is largely well put together and likely to promote new thinking and research, and also places this work in context and explains why it matters (not least in contributing to our understanding of how and why plants use silicon, and linking global C and Si cycles, and the role of plants in the regulation of both over geological timescales).

Thank you for your insightful comments and suggestions regarding our manuscript. We appreciate your recognition of the novelty and potential impact of our hypothesis.

2. The hypothesis is somewhat supported, but the final sentence of the paper is perhaps more accurate than the title – evolution of Si in plants correlates with climate, but whether that is temperature or water/VPD driven remains unclear. There are data gaps in the global data sets (most data from mid-latitudes). The authors note (L32), some patterns found could be linked to high transpiration rather than high temperatures, and the manuscript would be substantially strengthened if this was better explored. ‘Transpiration’ is mentioned just once in the manuscript. Considering vapour pressure deficit (VPD) seems the most straight forward way to explore if managing water loss or temperature stress is the key driver here, but is currently missing from the

manuscript. It may not be possible to retrospectively measure this for the field data collections and experiments, but adding it to global distribution analysis should be possible.

We agree transpiration rate is a significant factor affecting plant Si concentration, so it should be included in our analyses. Your suggestion is similar to those of earlier reviewers, so this comment has already been addressed, but to reiterate, we have now added eight additional climate variables to our analysis of the field samples. These are vapor pressure difference (VPD), average ground surface temperature (GST), evaporation (EVP), precipitation (PRE), average pressure (PRS), average relative humidity (RHU), average wind speed (WIN), and surface downwelling shortwave flux in air (RSDS). In addition, we have added new experiments to investigate the effect of temperature on Si accumulation in rice under the same humidity. The results of field sampling indicate that MAT is the most important climate variable affecting phytolith concentration in high-Si plant leaves, while VPD is not (new Fig. 3c-f). The results of the control experiment also proved that, at the same humidity (70%), a 2°C increase in temperature (26 °C vs 28 °C) for 10 d of incubation resulted in a 37% increase in leaf Si concentration of rice (new Fig S3). These analyses and experiments helped us eliminate the influence of water/VPD. In addition, we have included a detailed discussion of these factors, in conjunction with the literature (lines 306-315, 358-367).

3. There is a need to better justify or clarify some of the methods also.

We have carefully reviewed the manuscript and provided additional details and explanations where necessary to ensure that our methodological approach is transparent and well-supported (see earlier responses for more details).

4. Main issues

It is not always clear what aspects of plant Si has been measured. Leaf Si or plant Si? Si concentration or phytolith concentration. At times the data sets used seems too high for whole plant Si, but if this is leaf Si, then it could be presenting only part of the story. This warrants discussion.

Thank you for your question. Due to the integration of research from multiple data sets -

from field collections, experiments and global datasets, there may have been some confusion in the data description. Overall, our research focuses on the Si concentration in the leaves. For historical evolution, we used leaf Si concentration data from de Tombeur (2023) published in *Trends in Ecology & Evolution*²² (see details in Table S7). For rice and wheat collected in China, we used data on leaf phytolith concentrations. There is a significant correlation between the Si concentrations and the phytolith concentrations of crop leaves²³⁻²⁵, because research has shown that phytoliths are the main form of Si in higher plants, accounting for more than 90% of total Si in plants²⁶. However, for non-Si accumulating plants such as weeping willow and winter jasmine, due to their low Si concentration, it is not possible to extract plant phytoliths. Therefore, we used HNO₃, H₂O₂, and HF (HF has the ability to dissolve silica) to digest leaves and used an inductively coupled plasma optical emission spectrometer (ICP-OES; PE-7000DV, USA) to determine Si concentration¹³. In order to provide readers with a better reading experience, we have provided more prominent and detailed descriptions of each data in the main text (lines 130-144, 194-197, 514-520) and legend of Fig. 3 and S7.

5. Stromberg et al's 2016 paper is referenced, but there is no mention of the idea presented in this work that the evolution of insect herbivores could be a driver of Si accumulation in plants. I appreciate this makes the argument in this paper less elegant, but I think it is important to note that others have suggested other possible drivers given the list presented here. I was surprised not to see the inclusion of a discussion of C3 vs C4 plants, given the focus on temperature. Brightly et al. was mentioned (L325-7), but not the aspects of photosynthetic pathways. This could deepen the discussion – why might Si not be linked to this aspect of plant biology, but still driven by temperature?

Thank you for your suggestion. The articles by Strömberg and Brightly are crucial for expanding our understanding of the possible influencing factors of plant silicification which is why we have cited them. We agree that we need to explain how Si is linked to these aspects of biology and we have now included a fuller discussion regarding these possibilities in the revised discussion, including a consideration of the role of insect herbivory (lines 368-380). Our review of the literature regarding mammalian herbivory leads us to the conclusion that there is currently insufficient evidence that mammalian herbivory is a driving factor in the evolution of plant

silicification but the role of insect herbivory is less clear.

In addition, we also discussed the impact of photosynthetic pathways (lines 381-392). Due to the dominance of C4 grass in many heavily grazed habitats, some authors believe that C4 grasses evolved stronger mechanical defenses than C3 grasses through increased phytolith deposition, in response to extensive ungulate herbivory²⁷. Brightly et al. tested the classic hypothesis that high grazing pressure in tropical grasslands and savannas led to the evolution of higher silica levels in C4 grasses, but did not find any consistent difference in Si concentration between C3 and C4 grasses ($p = 0.22$) and thus rejected this hypothesis²⁸. In addition, the study also found that habitat types have no effect on plant silicon concentration ($p = 0.29$).

Our review of the literature has helped us exclude the effects of mammalian grazing, photosynthetic pathways, habitat types, and carbon dioxide concentration on plant Si accumulation over evolutionary time; we consider that our additional experiments and analysis, and our original findings, strongly support our viewpoint (lines 347-403). We sincerely thank the reviewer again for their suggestions, which have been very helpful in helping us supplement our arguments in our revised discussion, which we hope is now more comprehensive and more clearly supports our conclusion that the influence of temperature on plant silicon accumulation is more convincing compared to other factors.

6. The paper focuses on the idea that temperature (mean annual temperature) could be a driver of plant silicification – regionally, but also over evolutionary timescales. This can be related to water stress, no attempt has been made to exclude this possibility. It is mentioned, where it is noted that patterns linked to high temperature L32 “could also be attributed to increased transpiration accelerating passive Si absorption” then not considered for the rest of the analysis. Repeating analysis with VPD measures rather than MAT could be illuminating. If areas with high temperatures, but little drought stress (ie. more tropical areas) did not show high Si accumulation, then the argument would be stronger for transpiration as a driver, rather than MAT.

- Stronger signals were shown in wheat than rice, and wheat is more likely to have evolved in conditions with water stress than the water hungry rice.

- Drought can often be correlated with high temperatures, so correlations could be masking true drivers.

- Figure 2 shows that the areas experiencing water limitations are least well covered either by the families selected or the data available.

Also the recent paper by Cooke and Carey (2023), which provides meta-analysis support that Si increases water movement in plants suffering drought (and salinity) stress, which would support the arguments made here, warrants a mention, and further suggests a discussion of transpiration is needed.

Thank you for your suggestion. Other reviewers have raised the same issue so we have addressed these points in previous responses. To summarise, we have added an analysis of the relationship between nine climate variables (including mean annual temperature, vapor pressure difference, average ground surface temperature, evaporation, precipitation, average pressure, average relative humidity, average wind speed, and surface downwelling shortwave flux in air) and leaf Si concentration of plant from field sampling (new Fig. 3). It was found that mean annual temperature is the most important positive factor affecting Si concentration in the leaves of high-Si plant species. We also set up a new experiment to control humidity and only change temperature, to verify the promoting effect of temperature on plant Si absorption and distinguish it from the effects of humidity-driven changes in transpiration. It was found that under the same humidity, the accumulation of Si in rice leaves was significantly higher at 28 °C than at 26 °C (new Fig. S3).

The relationship between drought and plant Si concentration has been the subject of much research. For example, some studies have found that leaf Si concentration is higher under drought conditions^{29,30}. As such, there is a hypothesis that silicification might have evolved during arid periods. However, many laboratory and field experiments have shown that drought reduces the leaf Si concentration^{31,32}. In summary, there is no clear conclusion on the relationship between water availability and plant silicon content. In addition, our field sampling results also indicated that moisture-related climate variables such as precipitation, average relative humidity, and vapor pressure deficit did not contribute significantly to explaining phytolith concentrations in wheat and

rice leaves (new Fig. 3c-f), which does not support the hypothesis that drought promotes silicon accumulation in plants.

Similarly, although some studies have demonstrated the role of silicon in helping plants resist drought stress^{33,34}, others have found that Si mainly improves plant drought resistance by affecting soil water availability and soil water retention, rather than through physiological processes caused by plant absorption of Si^{35,36}. In contrast, the mechanism by which Si helps plants tolerate high temperatures often occurs within the plant body³⁷. Overall, compared to the role of temperature, the evidence for water availability as a driving factor for the evolution of plant silicification is relatively weak and we feel our revised discussion reflects that position whilst ensuring we fully consider alternative mechanisms (lines 358-367).

7.The authors have used multiple lines of evidence which is a strength, but this may mean there word count has been used to explain all of these, possibly the reason deeper consideration of some key issues around the main topic are missing, e.g. the potential role of transpiration. That said, I did really enjoy reading the discussion.

Thank you for your reminder. We agree the role of transpiration in the accumulation of silicon in leaves is very important and deserves a full discussion, so we have extended the section on transpiration in the revised manuscript (lines 306-315).

8.Specific comments

Figure 2 – y axis needs a scale, or perhaps two.

Thank you for your reminder. We have redrawn the figure.

9.Figure 3 – why phytolith concentration instead of silicon concentration? What proportion of total plant silicon is phytoliths? When considering functions, this is important.

Thank you for your question. We have already provided an answer earlier, and further explanation was provided here. In the analysis of Si concentration, it is generally believed that the extraction and determination of phytoliths are faster and more convenient than those of total Si, because the determination of total Si requires the use of hydrofluoric acid for digestion³⁸, which is

regulated and highly toxic, and may also corrode testing instruments. Research has shown that phytoliths are the main form of Si in higher plants, accounting for more than 90% of total Si in plants²⁶, and have a very close correlation with plant Si concentration^{23,24}. Therefore, for rice and wheat, phytoliths can reflect their silicon content. However, for non-Si accumulating plants such as weeping willow and winter jasmine, due to their low Si concentration, it is not possible to extract plant phytoliths. Therefore, we used HNO₃, H₂O₂, and HF to digest leaves and used an inductively coupled plasma optical emission spectrometer (ICP-OES; PE-7000DV, USA) to determine Si concentration¹³.

10. Figure 3d is not convincing, with an r^2 of 0.03. A high number of samples has led to the significance value rather than it representing a strong pattern. I worry that this means it reports a relationship between increased transpiration rather than Si use in hot climates. The reason rice does not show such a strong relationship is because it is semi-aquatic and transpiration rates are likely different compared to wheat.

Thank you for your reminder. We conducted additional analysis on upland rice and found a strong correlation result ($R^2=0.23$) that is similar to wheat (Fig. S5, lines 139-141). Therefore, we speculate that, as you suggest, the weak correlation we found with lowland rice may be caused by its aquatic environment. Temperature affects Si uptake, not only partly by its effect on energetically expensive Si transporters like Lsi2, but also through its effect on stomatal conductance and transpiration. We have added a discussion to better reflect the complex relationship between temperature, transpiration, and plant Si concentration in the main text (lines 306-315).

11. Figure 4 – This figure is really unclear. It does not link to the text well. It would be better to separate 4a from 4b&c, as the figure title is confusing. Red is used to represent both warm periods and high-Si species, which is problematic.

Thank you for your suggestion. We have made significant revisions to the figure, in response to your comment and those of the other reviewers (see earlier responses):

(i) We have separated 4a and 4b & c. The support provided by Figure 4a for our hypothesis is

not as important as 4b & c, so we moved it and the original sequence information to supplementary materials (new Fig. S6, Table S5 and S6). Here, we have redrawn and highlighted Fig. 4b and c, which strongly demonstrate the correlation between the appearance time of high-Si and low-Si plant families and Earth temperature.

(ii) To help readers better understand the relationship between the temperature thresholds and the evolution, we have annotated the corresponding geological period names and family names directly on the revised figure (new Fig. 4a).

(iii) For the figure of the phylogenetic tree, we have removed the red shading on the species names, as it was difficult to read and potentially confusing. Instead, we have directly added the leaf Si concentration values for each species next to their names on the phylogenetic tree (new Fig. S6).

12. Figure 4a - should be noted that all of these species are grasses. What are the implications of this? What is the value of including multiple species from within a genus, rather than looking at more genera?

Due to the high conservation of the Lsi series proteins, almost all homologous proteins produced by BLASTP search using the rice Lsi proteins belong to the Poaceae family. Therefore, genetic evolutionary analysis cannot provide us with supporting information beyond the Poaceae family. After discussion, we believe that it is thus more appropriate for this analysis to appear in the supplementary materials (e.g. Fig. S6) rather than in the main text.

13. Figure 4b – It is hard to link this to the next. List the families and label the climate periods perhaps?

Thank you for your suggestion. In the new version of the figure, information on the families and climate periods have been added.

14. Figure 4c – This figure is much more complicated than it needs to be. The curves on these which extend beyond the data, are not needed, nor the legend, which repeats the same information as the x axis. The box-plots with points (jitter) should be sufficient.

We have simplified the figure and hope it brings a better reading experience.

15.L78-79: This level of description (ie. defining high and low Si accumulation) is well done.

Thank you for your affirmation. In the new version of the manuscript, we have provided clear definitions of high and low temperatures based on the literature.

16.L85-87: Could consider tropics vs deserts etc here? The distribution of species seems quite dominated by some latitudes.

We compared the distribution ratio of high-Si plants and low-Si plants in tropical regions (-23.5° - 23.5°) and found that 14.2% of high-Si plants are distributed in the tropics, while only 13% of low-Si plants are. This result indicates that high-Si plants grow more in areas with higher temperatures than low-Si plants (lines 113-114).

17.L95: Significant-> significant but very weak, also why phytolith concentrations? I think from L33 onwards, this is actually Si concentration, but if no, why was something different measured, and what are the implications?

Based on your suggestion, we have changed the wording to “significant but very weak”. For wheat and rice, we measured phytoliths to quickly evaluate the silicon levels of many different individuals. Research^{23,24} has shown the strong correlation between phytoliths and total silicon, therefore this is an appropriate method to evaluate plant silicon content.

18.L123: include -> included.

It has been corrected.

19.L132-135: this sentences has multiple grammar errors

It has been corrected.

“Unlike the evolutionary information collected from different literature sources on the Angiosperm Phylogeny Website, the data from Li et al. (2019) was from their plastid phylogenomic tree, which was based on 80 genes from 2,881 plastid genomes and 62 fossil calibrations.”

20.L185: It is not clear how this paragraph relates to Figure 4, if at all. This Figure, and this paragraph and the one before could be much clearer and better joined. Could put some of the terms e.g. “Middle Eocene Thermal Maximum” on Figure 4b if they are part of the same data etc.

Thank you for your suggestion. Figure 4b depicts the occurrence of plant families with different leaf Si concentrations during warming and cooling episodes. Within the same family, there may be significant differences in Si concentration among plants from different sub-clades (e.g., subfamilies within a family), especially for families with a large number of species. In order to provide more evidence for the hypothesis that temperature drives the evolution of plant Si concentration, we selected the five families with the highest number of species among angiosperms (including Asteraceae, Orchidaceae, Fabaceae, Rubiaceae, and Poaceae) for separate analysis. The data here is not from Figure 4b, but from Table S10. By analyzing the relationship between the differences in Si concentration among different sub-clades within the same family and the historical temperature of the Earth, we still found that high-Si sub-clades are closely related to high temperature periods, while low-Si sub-clades are more likely to occur during low temperature periods. We have updated the description (lines 231-239) in the hope of providing greater clarity for readers.

21.L187: it seems odd to refer to the evolution of the climate when the paper is arguing that climate is an evolutionary driver of Si accumulation. Consider re-phrasing.

We have changed the wording to make it clearer (line 265-266).

“Our research suggests that temperature plays a significant role in plant Si accumulation.”

22.L198: Only vertebrate herbivores have been mentioned in the paper, despite Stromberg’s suggestion, so this should also clarify that statement only refers to vertebrate herbivores.

Thank you for your reminder. We have made this clarification.

23.L201: Yes I think the role of temperature can be stated so confidently, without exploration of discussion of water loss or water stress.

Thank you for your support. Based on previous results and newly supplemented experimental

data, our research results can prove that temperature plays a very important role in both current and historical periods. In addition, we have specifically discussed the inadequacy of evidence regarding the impact of other hypotheses on silicon accumulation, including humidity (or drought), herbivores, habitat types, photosynthetic pathways, and CO₂ levels (lines 347-403).

24.L267-8: This sentences need more details to be meaningful. Perhaps an example, otherwise it is too vague. The previous sentences refers to cold weather preventing evolution of high Si use, but this sentence talks about the ‘loss’ of high Si capacity. How are the two linked? Do species change Si accumulating capacity or do families that do or don’t accumulate Si flourish/expand at different times?

Thank you for your reminder. Our original intention here is to summarize the previous discussion and further discuss from the perspective of molecular biology the possible reasons why many plants do not utilize silicon, in order to highlight areas for future research. We agree with you that this argument is rather unclear and as we do not have convincing arguments about molecular reasons driving the loss of Si over evolutionary time, we have deleted this paragraph.

25.L275-6: “and our findings on the link between plant silicification and changes in the Earth's climate” – yes, I think there is strong evidence for this, but which aspect of climate hasn’t been really clearly pinned down.

Thank you for your suggestion. We have clarified the impact of temperature here.

References

1. Jones, L. H. P. & Handreck, K. A. in *Advances in Agronomy* (ed. Norman, A. G.) 107-149 (Academic Press, 1967).
2. Ma, J. F. & Takahashi, E. in *Soil, Fertilizer, and Plant Silicon Research in Japan* (eds. Ma, J. F., & Takahashi, E.) 63-71 (Elsevier Science, 2002).
3. Coskun, D. et al. The controversies of silicon's role in plant biology. *NEW PHYTOLOGIST* **221**, 67-85 (2019). <https://doi.org/10.1111/nph.15343>.
4. Hammer, K. & Khoshbakht, K. A domestication assessment of the big five plant families. *Genet. Resour. Crop Ev.* **62**, 665-689 (2015). <https://doi.org/10.1007/s10722-014-0186-2>.
5. Khoshbakht, K. & Hammer, K. Species Richness in Relation to the Presence of Crop Plants in Families of Higher Plants. *JOURNAL OF AGRICULTURE AND RURAL DEVELOPMENT IN THE TROPICS AND SUBTROPICS* **109**, 181-190 (2008).
6. Scotese, C. R., Song, H., Mills, B. J. W. & van der Meer, D. G. Phanerozoic paleotemperatures: The earth’ s changing climate during the last 540 million years. *Earth-Sci. Rev.* **215**, 103503

- (2021). <https://doi.org/10.1016/j.earscirev.2021.103503>.
7. Scotese, C. R. An Atlas of Phanerozoic Paleogeographic Maps: The Seas Come In and the Seas Go Out. *Annu. Rev. Earth Pl. Sc.* **49**, 679-728 (2021). <https://doi.org/10.1146/annurev-earth-081320-064052>.
 8. Peng, S. et al. Increased N-use efficiency using a chlorophyll meter on high-yielding irrigated rice. *Field Crop. Res.* **47**, 243-252 (1996). [https://doi.org/https://doi.org/10.1016/0378-4290\(96\)00018-4](https://doi.org/https://doi.org/10.1016/0378-4290(96)00018-4).
 9. Xiong, D. L. et al. SPAD-based leaf nitrogen estimation is impacted by environmental factors and crop leaf characteristics. *Sci. Rep.-UK* **5** (2015). <https://doi.org/10.1038/srep13389>.
 10. Shah, S. H., Houborg, R. & McCabe, M. F. Response of Chlorophyll, Carotenoid and SPAD-502 Measurement to Salinity and Nutrient Stress in Wheat (*Triticum aestivum* L.). *AGRONOMY-BASEL* **7** (2017). <https://doi.org/10.3390/agronomy7030061>.
 11. Wakiyama, Y. The Relationship between SPAD Values and Leaf Blade Chlorophyll Content throughout the Rice Development Cycle. *JARQ-Jpn. Agr. Res. Q.* **50**, 329-334 (2016). <https://doi.org/10.6090/jarq.50.329>.
 12. Hodson, M. J., White, P. J., Mead, A. & Broadley, M. R. Phylogenetic Variation in the Silicon Composition of Plants. *Ann. Bot.-London* **96**, 1027-1046 (2005). <https://doi.org/10.1093/aob/mci255>.
 13. Mitani-Ueno, N. et al. A silicon transporter gene required for healthy growth of rice on land. *Nat. Commun.* **14** (2023). <https://doi.org/10.1038/s41467-023-42180-y>.
 14. Huang, S. et al. The ZIP Transporter Family Member OsZIP9 Contributes To Root Zinc Uptake in Rice under Zinc-Limited Conditions^[OPEN]. *Plant Physiol.* **183**, 1224-1234 (2020). <https://doi.org/10.1104/pp.20.00125>.
 15. Fageria, N. K., Carvalho, G. D., Santos, A. B., Ferreira, E. & Knupp, A. M. Chemistry of Lowland Rice Soils and Nutrient Availability. *Commun. Soil Sci. Plan.* **42**, 1913-1933 (2011). <https://doi.org/10.1080/00103624.2011.591467>.
 16. Grigas, A., Savickas, D., Steponavicius, D., Niekis, Ä. & Balciunas, J. The Influence of Different Irrigation Scenarios on the Yield and Sustainability of Wheat Fodder under Hydroponic Conditions. *AGRONOMY-BASEL* **13** (2023). <https://doi.org/10.3390/agronomy13030860>.
 17. Jiang, D., Wu, H., Cai, H. & Chen, G. Silicon confers aluminium tolerance in rice via cell wall modification in the root transition zone. *Plant, Cell and Environment* (2022). <https://doi.org/10.1111/pce.14307>.
 18. Piernas, V. & Guiraud, J. P. Disinfection of rice seeds prior to sprouting. *J. Food Sci.* **62**, 611-615 (1997). <https://doi.org/10.1111/j.1365-2621.1997.tb04443.x>.
 19. Pang, Z. et al. Silicon-phosphorus pathway mitigates heavy metal stress by buffering rhizosphere acidification. *Sci. Total Environ.*, 166887 (2023). <https://doi.org/https://doi.org/10.1016/j.scitotenv.2023.166887>.
 20. Null, R. C. T. R. et al. R: A language and environment for statistical computing. *Computing* **1**, 12-21 (2011).
 21. Seifert, E. Origin Pro 9.1: Scientific Data Analysis and Graphing Software-Software Review. *J. Chem. Inf. Model.* **54**, 1552 (2014). <https://doi.org/10.1021/ci500161d>.
 22. de Tombeur, F. et al. Why do plants silicify? *Trends Ecol. Evol.* **38**, 275-288 (2023). <https://doi.org/10.1016/j.tree.2022.11.002>.
 23. Sun, X., Liu, Q., Tang, T., Chen, X. & Luo, X. Silicon Fertilizer Application Promotes Phytolith

- Accumulation in Rice Plants. *Front. Plant Sci.* **10** (2019). <https://doi.org/10.3389/fpls.2019.00425>.
24. Song, Z., Wang, H., Strong, P. J. & Guo, F. Phytolith carbon sequestration in China's croplands. *Eur. J. Agron.* **53**, 10-15 (2014). <https://doi.org/10.1016/j.eja.2013.11.004>.
 25. Yang, X. M. et al. Silicon in paddy fields: Benefits for rice production and the potential of rice phytoliths for biogeochemical carbon sequestration. *Sci. Total Environ.* **929** (2024). <https://doi.org/10.1016/j.scitotenv.2024.172497>.
 26. Song, Z., Müller, K. & Wang, H. Biogeochemical silicon cycle and carbon sequestration in agricultural ecosystems. *Earth-Sci. Rev.* **139**, 268-278 (2014). <https://doi.org/10.1016/j.earscirev.2014.09.009>.
 27. Bouchenak-Khelladi, Y. et al. The origins and diversification of C₄ grasses and savanna-adapted ungulates. *Global Change Biol.* **15**, 2397-2417 (2009). <https://doi.org/10.1111/j.1365-2486.2009.01860.x>.
 28. Brightly, W. H., Hartley, S. E., Osborne, C. P., Simpson, K. J. & Strömberg, C. A. E. High silicon concentrations in grasses are linked to environmental conditions and not associated with C₄ photosynthesis. *Global Change Biol.* **26**, 7128-7143 (2020). <https://doi.org/10.1111/gcb.15343>.
 29. Webb, E. A. & Longstaffe, F. J. Climatic influences on the oxygen isotopic composition of biogenic silica in prairie grass. *Geochim. Cosmochim. Ac.* **66**, 1891-1904 (2002). [https://doi.org/10.1016/S0016-7037\(02\)00822-0](https://doi.org/10.1016/S0016-7037(02)00822-0).
 30. Katz, O., Lev-Yadun, S. & Bar, P. Plasticity and variability in the patterns of phytolith formation in Asteraceae species along a large rainfall gradient in Israel. *Flora* **208**, 438-444 (2013). <https://doi.org/10.1016/j.flora.2013.07.005>.
 31. Wade, R. N., Donaldson, S. M., Karley, A. J., Johnson, S. N. & Hartley, S. E. Uptake of silicon in barley under contrasting drought regimes. *Plant Soil* **477**, 69-81 (2022). <https://doi.org/10.1007/s11104-022-05400-w>.
 32. Vandegeer, R. K. et al. Leaf silicification provides herbivore defence regardless of the extensive impacts of water stress. *Funct. Ecol.* **35**, 1200-1211 (2021). <https://doi.org/10.1111/1365-2435.13794>.
 33. Cooke, J. & Leishman, M. R. Consistent alleviation of abiotic stress with silicon addition: a meta-analysis. *Funct. Ecol.* **30**, 1340-1357 (2016). <https://doi.org/10.1111/1365-2435.12713>.
 34. Cooke, J. & Carey, J. C. Stress alters the role of silicon in controlling plant water movement. *Funct. Ecol.* **37**, 2985-2999 (2023). <https://doi.org/10.1111/1365-2435.14447>.
 35. Kuhla, J., Pausch, J. & Schaller, J. Effect on soil water availability, rather than silicon uptake by plants, explains the beneficial effect of silicon on rice during drought. *Plant, Cell & Environment* **44**, 3336-3346 (2021). <https://doi.org/10.1111/pce.14155>.
 36. Ryalls, J., Moore, B. D. & Johnson, S. N. Silicon uptake by a pasture grass experiencing simulated grazing is greatest under elevated precipitation. *BMC ECOLOGY* **18** (2018). <https://doi.org/10.1186/s12898-018-0208-6>.
 37. Saha, G., Mostofa, M. G., Rahman, M. M. & Tran, L. P. Silicon-mediated heat tolerance in higher plants: A mechanistic outlook. *Plant Physiol. Bioch.* **166**, 341-347 (2021). <https://doi.org/10.1016/j.plaphy.2021.05.051>.
 38. Feng, X. B., Wu, S. L., Wharmby, A. & Wittmeier, A. Microwave digestion of plant and grain standard reference materials in nitric and hydrofluoric acids for multi-elemental determination by inductively coupled plasma mass spectrometry. *J. Anal. Atom. Spectrom.* **14**, 939-946 (1999).

<https://doi.org/10.1039/a804683b>.

Response to Reviewers

Manuscript No.: NCOMMS-24-18614A

Title: Convergent evidence for the temperature-dependent emergence of silicification in terrestrial plants

Dear Reviewers,

Thank you for your efforts in reviewing our manuscript and helping us improve the quality of the paper. We greatly appreciate your professionalism and dedication. In response to your suggestions, we have made enhancements to the manuscript. We believe that these revisions provide more evidence and a better reading experience for the audience. The knowledge presented in this manuscript regarding the relationship between silicon-accumulating plants and temperature is valuable to a wider readership, contributing to a better understanding of plant silicon biology and its implications for global climate change.

Comments from Reviewers:

Reviewer #1:

I read your replies to my previous evaluation of your manuscript and appreciated that you have accepted the comments. After careful reading all your responses to other reviewers and the new version of the manuscript I add just a few additional notes. Generally, the text was considerably improved, and many new data were included. However, there are some points which I recommend clarifying or modify.

We appreciate your recognition of our previous manuscript. Thank you for your thorough and insightful review of our manuscript. We have made revisions based on the comments from you and the other reviewers.

1. The data about the effect of silicon in case of other stresses cannot be completely neglected. So, I cannot agree with the sentence:

Page 34 ...In addition, we have specifically discussed the inadequacy of evidence regarding the impact

of other hypotheses on silicon accumulation, including humidity (or drought), herbivores, habitat types, photosynthetic pathways, and CO₂ levels (lines 347-403).

You can discuss and support your opinion by your experiments, but the large amount of data showing the positive effect of Si on plants exposed to some other biotic or abiotic stresses cannot be neglected.

We fully agree with your viewpoint. Numerous studies have demonstrated the positive role of silicon in helping plants resist drought, nutrient deficiencies, lodging, diseases, and pests, which we also emphasized in the introduction (lines 3-6). At the suggestion of other reviewers, these discussions were added with the aim of stimulating more interesting research in the future by reviewing other factors that may affect the evolution of plant silicification.

2. The other problem may be the Si quantification by extraction of phytoliths and considering it as silicon concentration of leaves – or even plants. Total plant silicon is certainly not only phytoliths. Please note the publications showing Si – cuticular layer, silica deposition in sclerenchyma cells (of leaves and stems) in many species. I understand the difficulty with the method of qualifications, but it should be noted and underlined that this may affect the results (if some comparative measurements are not possible).

We commend your professional review opinions. Indeed, phytoliths are not the only form of silicon present in plants. But research has shown that phytoliths are the main form of silicon deposition in higher plants, accounting for over 90% of the total silicon in plants¹. In addition, there is a significant positive correlation between the phytolith concentration and the silicon concentration in plant leaves ($R^2 > 0.98, p < 0.01$)^{2,3}. Therefore, the use of phytoliths to reflect experimental results is reliable in our study.

3. And when speaking about Si in plants, do not forget about deposits of silica in roots. The data about root Si content or concentration are scarce.

As you said, in previous studies, more attention has been paid to leaf silicon, while there is relatively less data on root silicon. In addition, several classic root silicon transporters can only affect shoot silicon concentration, but cannot affect root silicon concentration⁴. Therefore, there are still many unknown molecular mechanisms that affect root silicon deposition. Your question is the direction for

future research.

4. Fig S2 fresh weight – of what – shoot + root?

The fresh weight in Fig S2 represents the total fresh weight of the entire plant (the sum of shoots and roots). We have clarified this in the legend. In addition, for the consistency of the paper, we also used this wording in Figure 1d.

Reviewer #2:

Thank you for the opportunity to conduct a review this revised manuscript. While the results and conclusions remain relatively unchanged, the readers confidence in the results is vastly improved by the changes made to the manuscripts by its authors. I appreciate the work that the authors did to address my previous comments and make improvements to the manuscript. In particular, I appreciate the analysis of non-silica accumulating plants across the temperature gradient (Fig. 3b), the inclusion of more predictors in that analysis, and the additional experiment teasing apart the effects of humidity from temperature on silica accumulation. These improvements to the manuscript provide great insight into the mechanism of leaf silification, how it can protect a plant against high temperatures, and clarity about its evolutionary origin. So congratulations on a paper that shows the best evidence to date that global temperatures can explain the evolution of silification across plant families.

Your comment is touching. We hope that researchers around the world can make greater progress in the field of plant silicification in the future.

1. A few additional questions and comments:

Pg 6, L 140: you report that many elements were measured in the plant leaves but apparently not carbon or nitrogen. Is that correct? As these are percentages, it seems pertinent to a mechanistic understanding of silica accumulation to know how carbon is changing as well. For example, it would be interesting to know if the background carbon is staying about the same and silica is increasing or if carbon is going down while silica goes up. Obviously, this is not critical to the dataset, but if you have the data on C and N I think it would be very interesting to know from a physiological standpoint.

Thank you for your suggestion. Although these elements do not provide strong support for our

viewpoint, studying their changes is still interesting and could be considered in our future research.

2. Pg 7, L 155: I think you mean “1,000”, right?

Yes. It has been changed to 1,000.

3. Pg 7, L 160-161: I don't understand the point of this sentence. Are you comparing high-Si and low-Si plants in the tropics? If so, 14.2% and 13% just don't seem that different and well within the expected range of error. Was this a typo and it was supposed to be 1.3%?

We have removed it.

4. Pg 8, Fig. 2: can you tell the reader what the blue and red dots are and their associated lines. Are these means and standard errors or something else? Also, can you add standard deviations to the table in C?

Thank you for your suggestion. We have redrawn the figure, and explained the meanings of dots and lines in the legend.

5. Pg 9, L 191-198: you refer to mean annual temperature (MAT) and average pressure (PRS) but not means of any of the variables (like PRE, GST, EVP, etc.). Why not? Were they not means or was this just an oversight in the rewriting of this section? Please clarify how the aggregation of these variables over time differs (or does not).

Thank you for your reminder. These variables are all means. We have updated the wording.

6. Pg 12, L 260-262: this sentence is deceptive. Are 59% and 54% somehow statically different? You did a nice job above describing the evolution of the reported proteins among families, but I am wondering if simply reporting the percentages is enough. I am wondering if there is a way to create a null model based on the tree and times since divergence and then randomize the families across the origin points many times to get the expected null distribution. Then you can ask is the observed values fall outside the 95% CI for the mean of that distribution. Without some sort of null model approach, I am not sure these percentages provide much insight.

We have removed these confusing percentages. This section further supports the view that high silicon

and low silicon plants are more inclined to evolve during high and low temperature periods on Earth, respectively. Readers can directly see specific information in the supplementary materials.

7. Pg 15, L 338: I suggest you change the sentence to read "...lignin, and therefore may damage..."

Suggestion taken.

8. Pg 15, 341-343: are "deep undercooling" and "deep supercooling" different? If so, explain. If not use the same phrase.

They have been unified as "deep supercooling".

9. Pg 17, L 392-398: I suggest you strengthen your wording here and say that while there is evidence that other factors adjust plant Si through plasticity the current analysis provides the strongest evidence to date that global temperatures provide an evolutionary explanation for the rise of plant silification. Don't shy away from your results and strong evidence – showcase them!

Thank you for your suggestion. We have emphasized the findings of this research in the last paragraph.

10. Pg 17, 401-410: same comment here. You spend the entire paper building a really strong case and then you cite a bunch of correlative studies that tended to focus on one or few factors. Citing paper with different results is important but don't do all that work and then say that you may be wrong and silification could have evolved for X, Y, and Z reasons.

Thank you for your suggestion. Discussing other possible factors is meaningful for readers to understand the progress in this field and to stimulate future research.

Reviewer #3:

This manuscript explores the idea that high air temperatures drove the evolution of silicon use by plants – that is, that plants use Si in hotter climates, in part to manage stresses associated with heat. The authors use multiple data sets – from field collections, experiments and global datasets to test the hypothesis. It is a novel perspective, and a provocative idea. The discussion is well put together and likely to promote new thinking and research, and also places this work in context and explains why it matters (not least in contributing to our understanding of how and why plants use silicon, and linking

global C and Si cycles, and the role of plants in the regulation of both over geological timescales). Substantial evidence is provided to support the hypothesis, but other possibilities and limitations are well explained. The careful consideration of the consistent reviewers comments has strengthened the manuscript considerably, and I found the arguments clear and convincing now.

Thank you for your recognition. Plant silicification is of great significance in biogeochemical cycles. Researchers from different disciplines need to work together to achieve more brilliant breakthroughs in the future.

1. The data used for Figure 2 comes from Hodson's data which was placed on a relative scale. If the raw data is plotted (Figure attached shows relative Si and raw Si data comparison) the high and low clades change. I think it would be worth repeating the analysis using the ordering from the raw data. The pattern may better support the hypothesis?

Thank you for providing valuable data. They are very helpful for this study. By comparing the two sets of data, we found that there was no change in the ranking of plant silicon content, so it will not affect the high silicon and low silicon clades selected in this study.

2. I have a few other small recommendations:

L1: for temperature-dependent -> for the temperature-dependent

Suggestion taken.

3. L155: 1.000 -> 1,000

Suggestion taken.

4. L161: only 13% -> 13% (I think if it was 40% compared to 13% you could use 'only' but a difference of 1.2 percentage points is not huge).

We have removed this sentence based on the suggestion of another reviewer.

5. L264: GAT needs an explanation.

It has been added in the legend.

6. L318: Delete: typical

Deleted.

7. L328: cell wall -> cell walls

Corrected.

8. L350: When analyzing the mechanism by which temperature affects plant Si uptake, it is usually disturbed by humidity, -> When analyzing the mechanism by which temperature affects plant Si uptake, it is usually confounded by effects of humidity,

Suggestion taken.

9. L351: “as temperature not only affects energetically expensive Si transporters like Lsi2” I don’t understand the logic of including this sentence, and how it fits with the rest of the argument.

Here, we discussed the possible physiological mechanisms by which temperature affects plant silicon content. On the one hand, the silicon transporter Lsi2 is energy consuming, and an increase in temperature within a certain range will enhance its activity. On the other hand, the increase in temperature promotes stomatal conductance and transpiration, which also play an important role in plant silicon absorption and transport. However, stomatal conductance and transpiration are also affected by humidity. This is why the mechanism by which temperature affects plant silicon absorption is often confused by the influence of humidity.

10. L350-353: This is a long and confusing sentence, and how it fits with the next sentence isn’t completely clear either. It would be worth re-writing as it is a key argument, possibly several arguments.

We have reorganized the sentence.

11. L373-375: References needed to support this claim.

Thank you for your reminder. This conclusion comes from the results of our study, and the content of that paragraph explains this conclusion.

12. L375-378: These are not plants

Thank you for your reminder. There is still controversy over whether algae can be studied as a part of botany^{5,6}. Therefore, we have corrected the wording and directly used algae for description.

13. L401: Not ideal to start a paragraph with 'for example'.

We started this paragraph with "One hypothesis suggests that".

14. Figure 3 b, c, d: the writing on the axis titles is very small. It might be better to have a longer figure that is legible. Figure 3 c, d, e, f: Please put parameters in the same order.

We have redrawn Figure 3 based on your suggestion.

15. Figure S6. Arrows have been added where they fall in warm periods, I don't think they are show where that is not the case. I think it would be fairer to show all arrows, at let the reader decided how convincing the results are. I think it will be a figure that still contributes support, but is more transparent in method.

Thank you for your suggestion. We labeled the silicon absorption and transport protein differentiation stages of species with higher silicon concentration in the figure to better express our viewpoint. In addition, readers can already obtain information about any node from Figure S6. Adding all arrows may cause the figure to become cluttered and unfocused.

References

1. Song, Z., Müller, K. & Wang, H. Biogeochemical silicon cycle and carbon sequestration in agricultural ecosystems. *Earth-Sci. Rev.* **139**, 268-278 (2014).
2. Yang, X. et al. Silicon in paddy fields: Benefits for rice production and the potential of rice phytoliths for biogeochemical carbon sequestration. *Sci. Total Environ.* **929**, 172497 (2024).
3. Song, Z., Wang, H., Strong, P. J. & Guo, F. Phytolith carbon sequestration in China's croplands. *Eur. J. Agron.* **53**, 10-15 (2014).
4. Ma, J. F., Tamai, K., Ichii, M. & Wu, G. F. A Rice Mutant Defective in Si Uptake. *Plant Physiol.* **130**, 2111-2117 (2002).
5. Sanders, W. B. The photoaerogens: algae and plants reunited conceptually. *Am. J. Bot.* **109**, 363-365 (2022).
6. Nature Photoaerogens? *Nat. Plants.* **8**, 317 (2022).